# Orthodenticle homeobox 2 is transported to lysosomes by nuclear budding vesicles

Jun Woo Park [1], Eun Jung Lee[1], Eunyoung Moon[2], Hong-Lim Kim[3], In-Beom Kim [3], Didier Hodzic[4], Namsuk Kim [1,5], Hee-Seok Kweon [2] & Jin Woo Kim [1] ✉

Transcription factors (TFs) are transported from the cytoplasm to the nucleus and disappear from the nucleus after they regulate gene expression. Here, we discover an unconventional nuclear export of the TF, orthodenticle homeobox 2 (OTX2), in nuclear budding vesicles, which transport OTX2 to the lysosome. We further find that torsin1a (Tor1a) is responsible for scission of the inner nuclear vesicle, which captures OTX2 using the LINC complex. Consistent with this, in cells expressing an ATPase-inactive Tor1aΔE mutant and the LINC (linker of nucleoskeleton and cytoskeleton) breaker KASH2, OTX2 accumulated and formed aggregates in the nucleus. Consequently, in the mice expressing Tor1aΔE and KASH2, OTX2 could not be secreted from the choroid plexus for transfer to the visual cortex, leading to failed development of parvalbumin neurons and reduced visual acuity. Together, our results suggest that unconventional nuclear egress and secretion of OTX2 are necessary not only to induce functional changes in recipient cells but also to prevent aggregation in donor cells.

Given the spatial segregation of chromosomes and ribosomes—the sites of transcription and translation of RNA, respectively—eukaryotic proteins that function in the nucleus must be transported to the nucleus after translation of their mRNA by ribosomes in the cytoplasm. Nuclear localization signals (NLSs), present in the majority of nuclear proteins, are recognized by the importin complex, which moves proteins to the nucleus through the nuclear pore complex (NPC)[1–3]. Conversely, nuclear export signals (NESs), which are responsible for cytoplasmic transport by the exportin complex[4,5], have also been identified in many nuclear proteins. However, NES sequences are less conserved than NLS sequences in nuclear proteins. This suggests that nuclear proteins lacking an NES are transported to the cytoplasm via alternative export pathways[6–8].

Several NES-independent nuclear export pathways have been identified[9–11], most of which remodel the nuclear envelope—a double-membrane structure continuous with the endoplasmic reticulum (ER)[10,11]. In one such pathway, nuclear components are trapped in the inner nuclear membrane, which forms vesicular intermediates in the internuclear membrane space before fusing with the outer nuclear membrane and releasing the nuclear components into the cytoplasm. This vesicular nuclear egress phenomenon underlies the cytoplasmic transport of herpes simplex virus (HSV) capsid particles[11,12] and *Par3* mRNA-containing ribonucleoproteins (RNPs)[10], which are larger than the diameter of the NPC (~39 nm) and thus cannot be transported through the pore[13]. In the latter case, torsin-1a (TOR1A), an AAA + ATPase localized to the internuclear membrane space, is responsible for scission of the inner nuclear membrane that captured *par3*-RNPs[14]. Thus, as demonstrated in *Drosophila*, the RNPs cannot be transported to the cytoplasm in cells lacking *torsin* or expressing an ATPase-inactive *torsin* mutant[14]. However, whether this is a general phenomenon that can be applied to other nuclear components is still unknown.

[1]Department of Biological Sciences and Stem Cell Research Center, Korea Advanced Institute of Science and Technology (KAIST), Daejeon 34141, South Korea. [2]Electron Microscopy Research Center, Korea Basic Science Institute, Cheongju 28119, South Korea. [3]Integrative Research Support Center, College of Medicine, The Catholic University of Korea, Seoul 06591, South Korea. [4]Department of Developmental Biology, Washington University School of Medicine, St Louis, MO 63110, USA. [5]Present address: Neurovascular Unit, Korea Brain Research Institute, Daegu 41062, South Korea. ✉e-mail: jinwookim@kaist.ac.kr

After export to the cytoplasm, nuclear proteins return to the nucleus or are degraded by cytoplasmic proteolytic machineries, such as the ubiquitin-proteasome system (UPS) and the lysosomal proteolysis that mediates endosomal and autophagosomal protein entrapment[15]. Otherwise, nuclear proteins are eliminated in the nucleus by nuclear proteolytic degradation pathways[16–18]. Given spatial limitations of the nucleus, defects in degradation or export of nuclear proteins could therefore result in the accumulation of proteins, which often form aggregates and cause proteotoxicity[19].

Homeodomain-containing proteins (HPs) are TFs that are transported to the nucleus to regulate the expression of genes involved in the development of organisms[20,21]. HPs have also been identified to participate in cytoplasmic events such as mRNA translation[22–25] and mitochondrial ATP synthesis[26,27]. Surprisingly, more than a dozen HPs, which possess cytoplasmic functions, are not synthesized in the cells in which they function, but instead are taken up from neighboring cells by intercellular protein transfer[28,29], a process identified as a general feature of HPs[30]. This unconventional trafficking of HPs requires secretion and penetration signal sequences in the homeodomain[31] and is also supported by hydrophobic amino acids outside the homeodomain[30]. However, many questions regarding the mechanism of HP secretion remain unanswered, whereas mechanisms by which HPs penetrate cells following binding to the glycosaminoglycan (GAG) sugar chains of proteoglycans have been identified[25,32,33].

In this study, we report that secretion of orthodenticle homeobox 2 (OTX2) starts with OTX2 binding to the LINC complex in the nuclear envelope. We also found that TOR1A in the internuclear membrane space and dynamin (DNM) in the cytoplasm involves in the scission of double-layered nuclear budding vesicles, which subsequently transport OTX2 to the lysosome (for degradation) and then to the extracellular space (for intercellular transfer). Moreover, secretion of Otx2 from cells in the choroid plexus (ChP) and subsequent transfer to parvalbumin (PV) neurons in the mouse visual cortex is suppressed in *Tor1a^AE* mice, which expresses an ATPase-inactive Tor1aΔE mutant, as well as in *LSL-KASH2;FoxJ1-CreER* mice, in which EGFP-KASH2 expressed in the internuclear membrane space of ChP cells to interfere with the formation of the LINC complex. Consequently, excess Otx2 remains in the nuclei of these mouse ChP cells and forms aggregates, resulting in diminished delivery of OTX2 to the visual cortex and delayed visual maturation.

## Results

### Localization of HPs in cytoplasmic puncta

A majority of HPs were previously shown to be secreted from cells and penetrate neighboring cells[30]. Interestingly, immunostaining has revealed punctate signals for secretory HPs (sHPs), including OTX2, EN2 and VAX1, in the cytoplasm of HeLa cells, in addition to their nuclear signals (Fig. 1a)[30]. In contrast, cytoplasmic HP puncta were hardly observed in HeLa cells expressing non-secretory HPs (nsHPs)[30], such as SHOX2 and HOXD4 (Fig. 1a), revealing a positive relationship between cytoplasmic puncta and HP secretion.

Given the potential ability of HPs to bind RNA[22,34], cytoplasmic HP puncta could reflect the presence HPs in RNP complexes. Thus, we examined whether cytoplasmic puncta are stress granules, which are composed of RNP complexes[35]. However, OTX2 cytoplasmic puncta did not overlap with Ras-GAP SH3 domain-binding protein (G3BP; Fig. 1b, left), a marker for stress granules[36]. OTX2 puncta were also not co-localized with synuclein-α (SNCA; Fig. 1b, right), a marker for cytoplasmic protein aggregates[37], indicating that they are neither insoluble protein aggregates. Consistent with this, cytoplasmic OTX2 was detectable only in detergent-soluble fraction, whereas nuclear OTX2 could be detectable in the insoluble fraction containing chromosomal histone 3A (H3A) as well as the soluble fraction (Fig. 1c).

## Vesicular identity of cytoplasmic OTX2 puncta

We further investigated the physical properties of cytoplasmic OTX2 by sucrose density gradient fractionation. OTX2 was detectable in sucrose gradient fractions #7 to #9 (Fig. 1d, e). These fractions did not contain green fluorescent protein (GFP), which is a soluble cytosolic protein detected in lighter fractions (#4 to #6). Instead, OTX2-positive fractions partially overlapped with the transmembrane protein, epidermal growth factor receptor 1 (EGFR1), and the caveolar protein, flotillin 1 (FLOT1) (Fig. 1d, e). Given that OTX2 binds to GAG sugar chains of proteoglycans[32,33], extracellular OTX2 could be co-fractionated by virtue of its association with transmembrane proteoglycans, such as the chondroitin sulfate proteoglycans (CSPG), neural/glial antigen 2 (NG2), and heparan sulfate proteoglycan (HSPG), syndecan 3 (SDC3) (Fig. 1d, e). To test this possibility, we treated cells with excess heparin to detach OTX2 from proteoglycans[30]. However, this did not change the distribution of OTX2 (Fig. 1d, e), suggesting that the co-fractionation of OTX2 with those proteoglycans is not likely attributable to direct interactions.

Next, we determined whether OTX2 localizes to intracellular membranous organelles by co-immunostaining with organelle-specific markers. We found that OTX2 immunoreactive signals in the cytoplasm were surrounded by those of lamin A and C (LMN-A/C), which comprise the nuclear lamina, but did not co-localize with translocase of outer membrane 20 (TOM20)-positive mitochondria (Fig. 1f [two leftmost columns], g). Cytoplasmic OTX2 was also co-localized with calnexin (CANX), an ER marker (Fig. 1f [third column from the left], g), but was not detected in the GM130-positive Golgi apparatus (Fig. 1f, [fourth column from the left], g). These results suggest that cytoplasmic OTX2 puncta could be the structures derived from the nucleus and/or ER.

Interestingly, OTX2 also co-localized with lysosomal membrane-associated protein 2 (LAMP2) (Fig. 1d–g). However, it was not detected in microtubule-associated protein 1A/1B light chain 3B (LC3B)-positive autophagosomes or CD63-positive late endosome/multivesicular bodies (MVBs) (Fig. 1f [two rightmost columns], g), which are known to entrap cytoplasmic proteins prior to fusion with lysosomes[38]. OTX2 was not detected in early endosome antigen 1 (EEA1)-positive endosomes, but was found in FLOT1-positive lipid rafts (Fig. 1f [fifth and sixth columns from the left], g). Together, these results suggest the possibility that OTX2 is transported in lipid raft-enriched membrane vesicles to the lysosome and/or the extracellular space without the mediation of endosomes, autophagosomes, or MVBs.

## Nuclear origin of OTX2 in the lysosome and extracellular space

We next examined whether the cytoplasmic puncta formation and secretion are correspondingly changed upon increasing cytoplasmic OTX2. To this end, we suppressed the nuclear import of OTX2 by replacing two arginine (R) residues in its NLS to alanine (A) (Supplementary Fig. 1a). The resultant OTX2(RR/AA) mutant was undetectable in the nucleus and instead was retained in the cytoplasm (Supplementary Fig. 1b, leftmost column). The cytoplasmic OTX2(RR/AA) overlapped strongly with mitochondria but sparsely with lysosomes (Supplementary Fig. 1b, two center columns). However, the OTX2(RR/AA) signals were not associated with autophagosomes (Supplementary Fig. 1b, rightmost column), which can engulf mitochondria[39].

Interestingly, cytoplasmic OTX2(RR/AA) was isolated in the insoluble fraction at a significant level (Supplementary Fig. 1c,d), whereas cytoplasmic wild-type (WT) OTX2 remained only in the soluble fraction (Fig. 1c). Notably, extracellular OTX2(RR/AA) levels were decreased compared with those of OTX2(WT) (Supplementary Fig. 1e, f). These results suggest that nuclear translocation of OTX2 is necessary for enhancing its solubility, lysosomal transport, and secretion into the extracellular space.

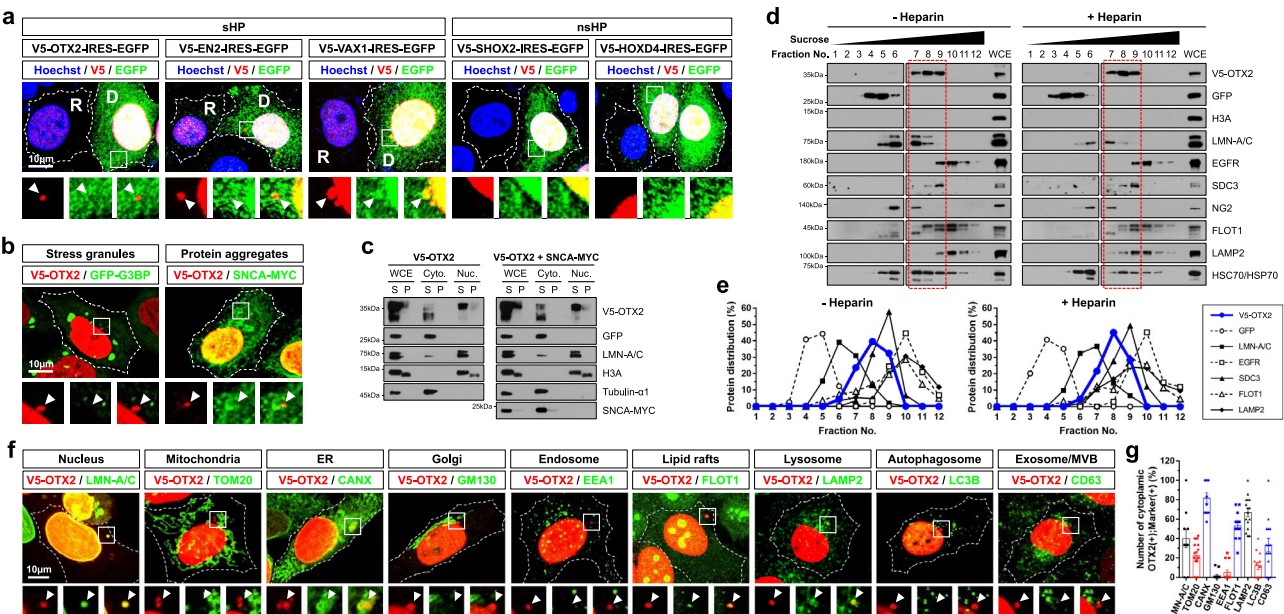

**Fig. 1 | Cytoplasmic OTX2 is located in the membrane fraction. a** V5-tagged human HPs, OTX2, EN2, VAX1, SHOX2, or HOXD4, overexpressed in HeLa cells were detected by immunostaining with anti-V5 antibody. GFP, which was expressed in the same transcripts with these HPs, was also co-stained with anti-GFP antibody. Nuclei of the cells were visualized by staining with Hoechst-33332 dye (Hoechst). These HPs were secreted from the GFP-positive donor (D) cells and transferred to the recipient (R) cells, which exhibit HPs without GFP. Dotted lines are the edges of the cells. Images in the bottom row are magnified versions of the boxed areas in the top row. Arrowheads point perinuclear and cytoplasmic HP puncta. Similar observations were made in 3 independent experiments, and representative images are shown. **b** V5-OTX2 and GFP-G3BP or SNCA-MYC overexpressed in HeLa cells were visualized by immunostaining with anti-V5 and anti-GFP or anti-MYC antibodies. Dense signals of GFP-G3BP and SNCA-MYC in the cytoplasm are stress granules and protein aggregates, respectively. Arrowheads indicate perinuclear and cytoplasmic V5-OTX2 puncta. Similar observations were made in 3 independent experiments, and representative images are shown. **c** Cytoplasmic (Cyto.) and nuclear (Nuc.) fractions of HeLa cells that overexpress GFP and V5-OTX2 with or without SNCA-MYC were separate as described in the Methods. The resultant fractions were incubated in RIPA buffer (PBS with 0.1% (final) SDS) to isolate soluble

supernatant (S) and insoluble pellet (P) by centrifugation. The cells were also lysed in RIPA buffer without subcellular fractionation to obtain the whole cell extracts (WCE). The proteins of our interest in each fraction were detected by western blotting (WB). Molecular weights of those proteins are shown in corresponding WB images. Similar results were obtained in 3 independent experiments, and representative blots are shown. **d** OTX2-HeLa cells were incubated with or without heparin (10 mg/ml for 3 h) prior to the separation into cytoplasmic and nuclear fractions. Cytoplasmic fractions (90% of total lysates) of HeLa cells that co-express GFP and V5-OTX2 were further separated by sucrose density gradient centrifugation. Proteins in 5% of each fraction and 0.2% of WCE were detected by WB. Dotted red boxes indicate OTX2-positive fractions. Similar results were obtained in 3 independent experiments, and representative blots are shown. **e** Distribution of proteins in each fraction was plotted in the graphs. **f** HeLa cells overexpressing V5-OTX2 were stained with antibodies that recognize V5 and organelle-specific markers. Arrowheads indicate perinuclear and cytoplasmic V5-OTX2 puncta. **g** The graph indicates the number of cytoplasmic OTX2 puncta co-localized with organelle markers in each cell. The columns represent means, and error bars denote standard error of the means (SEM) obtained from 4 independent experiments. Source data are provided as a Source Data file.

## Transport of OTX2 from the nucleus to the lysosome for degradation and secretion

Given the enrichment of hydrolases in the lysosomes, we next tested whether OTX2 is transported to the lysosome for degradation. We found the levels of OTX2 were elevated significantly in cells treated with bafilomycin A1 (BafA1), an inhibitor of the lysosomal proton pump[40], or chloroquine (CQ), a chelator of lysosomal protons[41], but were unaffected by MG-132, an inhibitor of the proteasome[42] (Fig. 2a, b and Supplementary Fig. 2a, b). The OTX2 elevation in BafA1-treated cells was mostly found in the fractions including LAMP2 (Supplementary Fig. 3). In keeping with these results, the number of cytoplasmic OTX2 puncta co-localized with LAMP2-positive lysosomes was also significantly increased in cells treated with BafA1 or CQ, but not by MG-132 (Fig. 2d, e and Supplementary Fig. 2c).

Interestingly, the amount of OTX2 in the extracellular space was also increased upon the inhibition of lysosomal function (Fig. 2a, c). Extracellular OTX2 was detected not only in the centrifugal supernatant (S100) fraction of growth medium containing insulin-like growth factor 1 (IGF-1), but also in the centrifugal precipitate (P100) fraction, which was enriched for membrane proteins like the IGF-1 receptor (IGF-1R) (Fig. 2f). OTX2 was elevated in both P100 and S100 fractions in the growth medium of cells treated with BafA1 (Fig. 2g, h). Given the absence of GFP in the growth media (Fig. 2f, g), the

extracellular OTX2 was not increased by non-specific release of intracellular proteins from dead cells. OTX2 secretion was also enhanced by treating the cells with lysosomal lipase inhibitors, such as imipramine (IMP)[43] and lalistat 1 (L1)[44] (Supplementary Fig. 4a, d). These inhibitors increased the cytoplasmic OTX2 level without changing the nuclear OTX2 level (Supplementary Fig. 4a–c). The excessive cytoplasmic OTX2 formed puncta that co-localized with LAMP2 (Supplementary Fig. 4e). These results suggest that OTX2 exhibits luminal and vesicular localization in the lysosome, which could fuse to the plasma membrane to release naked and vesicular OTX2, respectively, into the extracellular space.

## OTX2 is detectable in nuclear egress membrane vesicles

We next tracked OTX2 in the nucleus and cytoplasm in greater detail by transmission electron microscopic detection of immunostaining (iTEM) signals. V5-tagged OTX2 was not only present in the nuclei of HeLa cells; it was also detected in perinuclear and cytoplasmic spaces (Fig. 3a, b), consistent with the results of fractionation (Fig. 1c–e). Significant numbers of cytoplasmic OTX2 iTEM signals were detected in vesicles surrounded by single or double-membrane layers (Fig. 3c and Supplementary Fig. 5a, b). Moreover, OTX2 iTEM signals were also found in perinuclear buds continuous with the inner and outer nuclear membranes (Fig. 3a and Supplementary Fig. 5b), suggesting that OTX2

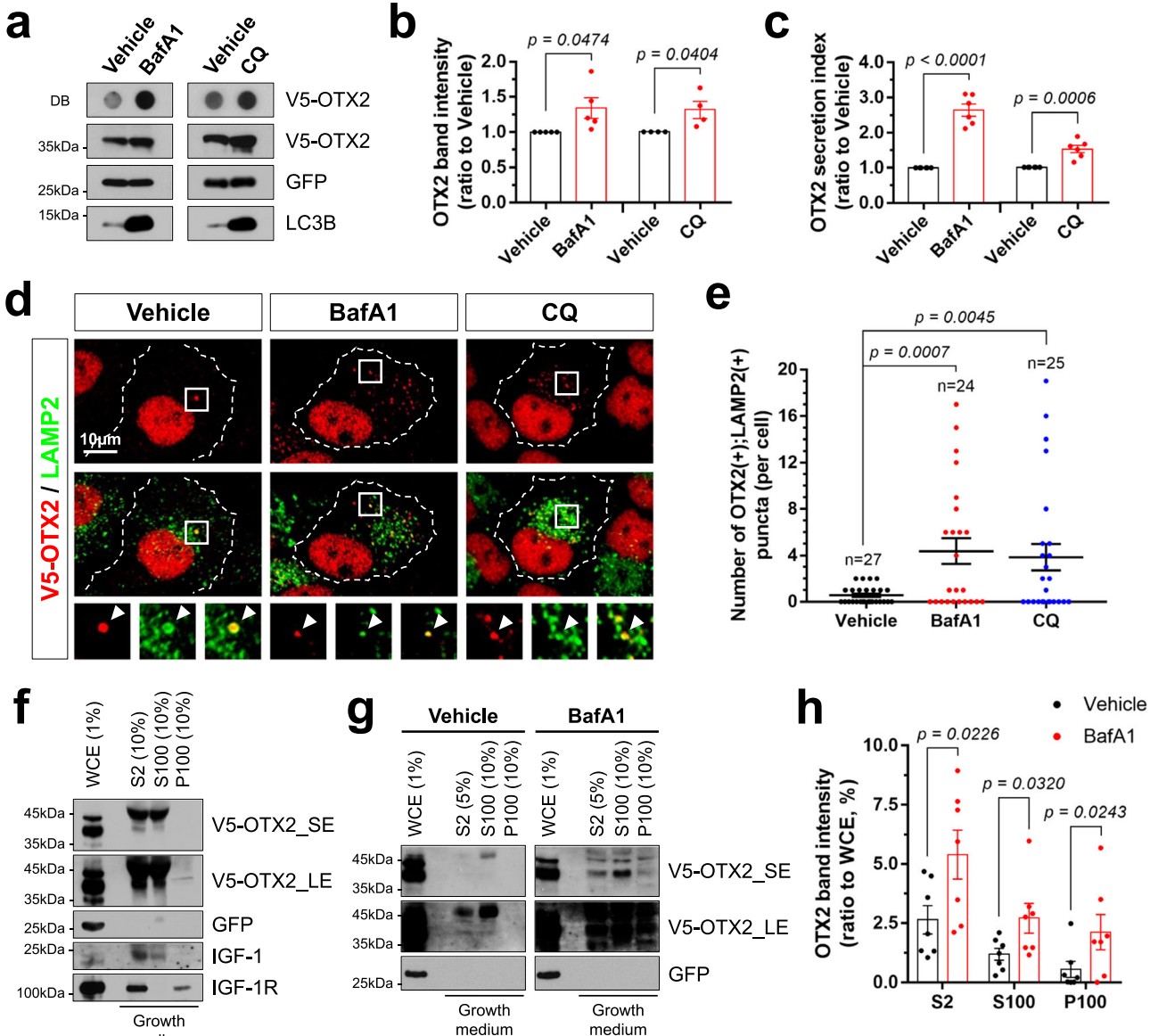

**Fig. 2 | OTX2 is accumulated in the lysosome and extracellular space upon lysosomal dysfunction. a** OTX2-HeLa cells were treated with lysosomal inhibitors, BafA1 (200 nM) and CQ (50 μM), or vehicle (DMSO) for 24 h. V5-OTX2 secreted into the growth medium of the cells was detected by dot blotting (DB) with an anti-V5 antibody (see details in Methods). Levels of V5-OTX2 and GFP in the cells were also detected by WB. Inhibition of lysosomal function by BafA1 and CQ was confirmed by detecting the accumulation of LC3B by WB. The graphs show relative intensity of WB bands of OTX2 (**b**) and relative secretion index of OTX2 determined by DB (**c**), respectively. The columns represent means, and error bars denote SEM. Results were obtained from 6 independent experiments. **d** OTX2-HeLa cells treated with vehicle, BafA1, or CQ were stained with antibodies that recognize V5 and LAMP2. Images in the bottom row are magnified versions of boxed areas in the top rows. Arrowheads indicate LAMP2-positive cytoplasmic puncta containing V5-OTX2.

**e** The number of OTX2(+);LAMP2(+) cytoplasmic puncta in each cell is shown by a dot in the graph. Long sidebars represent mean values and error bars denote SEM from 4 independent experiments. The numbers of cells analyzed are shown in the graph. **f** P100 fraction containing membranous structures was separated from the S100 fraction of OTX2-HeLa cell growth medium (see details in Methods). The proteins of interest in each fraction were then detected by WB. SE, short exposure (<10 min); LE, long exposure (>10 min) of WB membrane. Similar results were obtained in 3 independent experiments, and representative blots are shown.
**g** OTX2-HeLa cells were treated with vehicle or BafA1 for 24 h. Then, the levels of proteins in each fraction were analyzed by WB. **h** The graphs show the mean values of relative OTX2 intensity determined by WB. Error bars represent SEM obtained from 7 independent experiments. Statistical analysis was performed by two-tailed Student's $t$-test (in **b**, **c**, **e**, **h**). Source data are provided as a Source Data file.

undergoes nuclear egress via nuclear budding vesicles. This hypothesis was further assessed by comparing the distribution of nuclear markers. Our results showed that OTX2 staining signals were enwrapped by those of LMN-A/C, a nuclear matrix marker; lamin-associated protein 1 (LAP1), an inner nuclear membrane marker; and spectrin repeat-containing nuclear envelope protein 2 (SYNE2), an outer nuclear membrane marker (Fig. 3d).

We also investigated the presence of OTX2 in membrane vesicles in vivo. In the adult mouse brain, *Otx2* is expressed in the choroid plexus (ChP), the pineal gland, and the cerebellum[45] (Fig. 3e, top). The

ChP was reported to secrete Otx2 to the cerebrospinal fluid (CSF) for subsequent transfer to PV neurons in the primary visual cortex (V1)[46,47] (Fig. 3e, bottom). Thus, we examined the distribution of Otx2 in ChP cells by iTEM. In mouse ChP cells, in addition to the majority signals in the nucleus, minor Otx2 iTEM signals were detected in the cytoplasmic vesicles and mitochondria (Fig. 3f–h and Supplementary Fig. 6), which is similar to the distribution observed for V5-OTX2 in HeLa cells (Fig. 3a–c). These results suggest that nuclear OTX2 might be trapped in the inner nuclear membrane prior to it budding off to form single-layered vesicles that fuse to the outer nuclear membrane to release

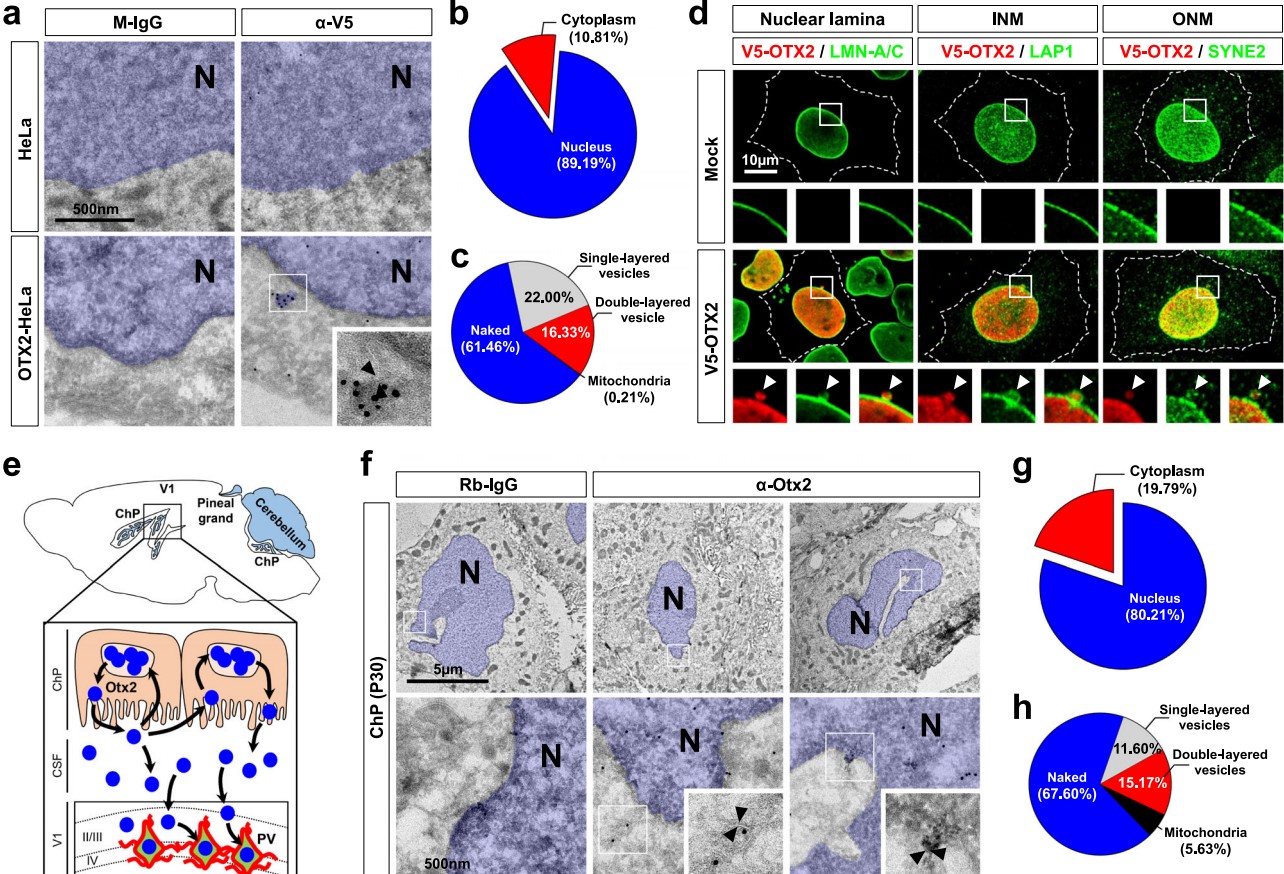

**Fig. 3 | OTX2 is detectable in double-layered membrane vesicles. a** Cryo-sections (50 nm) of HeLa and OTX2-HeLa cells were stained with gold-labeled anti-mouse IgG that detects preimmune mouse IgG and mouse anti-V5 antibody (α-V5) on the sections. The gold particles of anti-mouse IgG on the section are shown in black dots. The inset is magnified image of the box area. Nuclear plasm (N) is pseudo-colored in blue, and nuclear membranes are pointed by with dotted arrowheads. **b** Pie chart shows the relative distribution of gold particles in the nuclei and cytoplasm of OTX2-HeLa cells ($n = 22$, from 6 independent batches). **c** Pie chart shows the locations of gold particles in the cytoplasm of OTX2-HeLa cells. Naked, not associated with a membrane ($n = 22$, from 6 independent batches). **d** HeLa cells transfected with Mock or V5-OTX2 were visualized using antibodies against V5 (for OTX2), LMN-A/C (for nuclear lamina), LAP1 (for the INM), or SYNE2 (for the ONM).

Images in the bottom rows are magnified versions of the boxed areas in the top rows. Dotted lines are the edges of the cells. Arrowheads indicate the perinuclear OTX2 puncta wrapped by nuclear lamina, INM, or ONM. Similar observations were made in 3 independent experiments, and representative images are shown. **e** The diagram shows *Otx2* mRNA (blue) expression pattern in adult mouse brain (top) and the route of Otx2 protein transfer from the ChP to V1 via the CSF (bottom). **f** Cryo-immunogold staining images of Otx2 in the sections (50 nm) of P30 C57BL/6 J mouse ChP. **g** Pie chart shows distribution of gold particles detected in the nuclei and cytoplasm ($n = 6$). **h** Pie chart shows the locations of gold particles in the cytoplasm ($n = 6$). Quantified images in **c** and **h** are provide in Supplementary Figs. 5 and 6, respectively.

naked OTX2 in the cytoplasm, or to make double-layered vesicles together with the outer nuclear membrane.

## Nuclear egress of OTX2 is regulated by TOR1A and DNM

Vesicular trafficking were previously reported to be involved in the nuclear egress of HSV and *par3*-RNP[10,11]. This process requires TOR1A, an AAA + ATPase, which contributes to the scission of vesicles from nuclear and ER membranes that leads to the release of membrane vesicles into the internuclear membrane space and ER lumen, respectively[48,49]. We thus tested whether TOR1A also regulates the formation of OTX2-containing vesicles. To this end, we co-expressed TOR1A(WT) or an ATPase-inactive TOR1A(ΔE302/E303) mutant (hereafter, TOR1AΔE)[50] with OTX2 in HeLa cells and then assessed OTX2 secretion and formation of cytoplasmic OTX2 puncta. Over-expressed TOR1A(WT) was expected to enhance severing of the inner nuclear membrane, whereas TOR1AΔE was expected to increase the blebs in the inner nuclear membrane by inhibiting membrane scission[14,50]. In support of this, the number of cytoplasmic OTX2 puncta was increased by co-expression of TOR1A(WT) (Fig. 4a–c). Conversely, TOR1AΔE co-expression decreased OTX2

cytoplasmic puncta while increasing OTX2 in nuclear blebs, which are surrounded by LMN-A/C and LAP1 (Fig. 4a–c and Supplementary Fig. 7). Consequently, TOR1A(WT) increased the OTX2 level in the cytoplasm and extracellular space, whereas TOR1AΔE decreased these levels (Fig. 4d).

Given the presence of OTX2 in double-layered membrane vesicles (Fig. 3a, f), scission of the inner nuclear membrane by TOR1A should be coupled to the severing of the outer nuclear membrane. Dynamin (DNM) is involved in the formation of cytoplasmic vesicles from various membrane structures, including the ER, Golgi apparatus, and plasma membrane[51,52]. Thus, scission of the outer nuclear membrane, which surrounds the OTX2-containing inner nuclear membrane, could also be regulated by DNM. We tested this possibility by treating cells with the DNM inhibitors, Dynasore (DNSR)[53] and MiTMAB[54]. Remarkably, the number of LMN-A/C-positive inner and SYNE2-positive outer nuclear membrane blebs, which contained OTX2, were increased in cells treated with DNSR or MiTMAB, whereas the number of OTX2 cytoplasmic puncta was decreased (Fig. 4e–g). We also found that both agents suppressed the secretion of OTX2 (Fig. 4h). These results suggest that OTX2 might be trapped in double-layered nuclear membrane

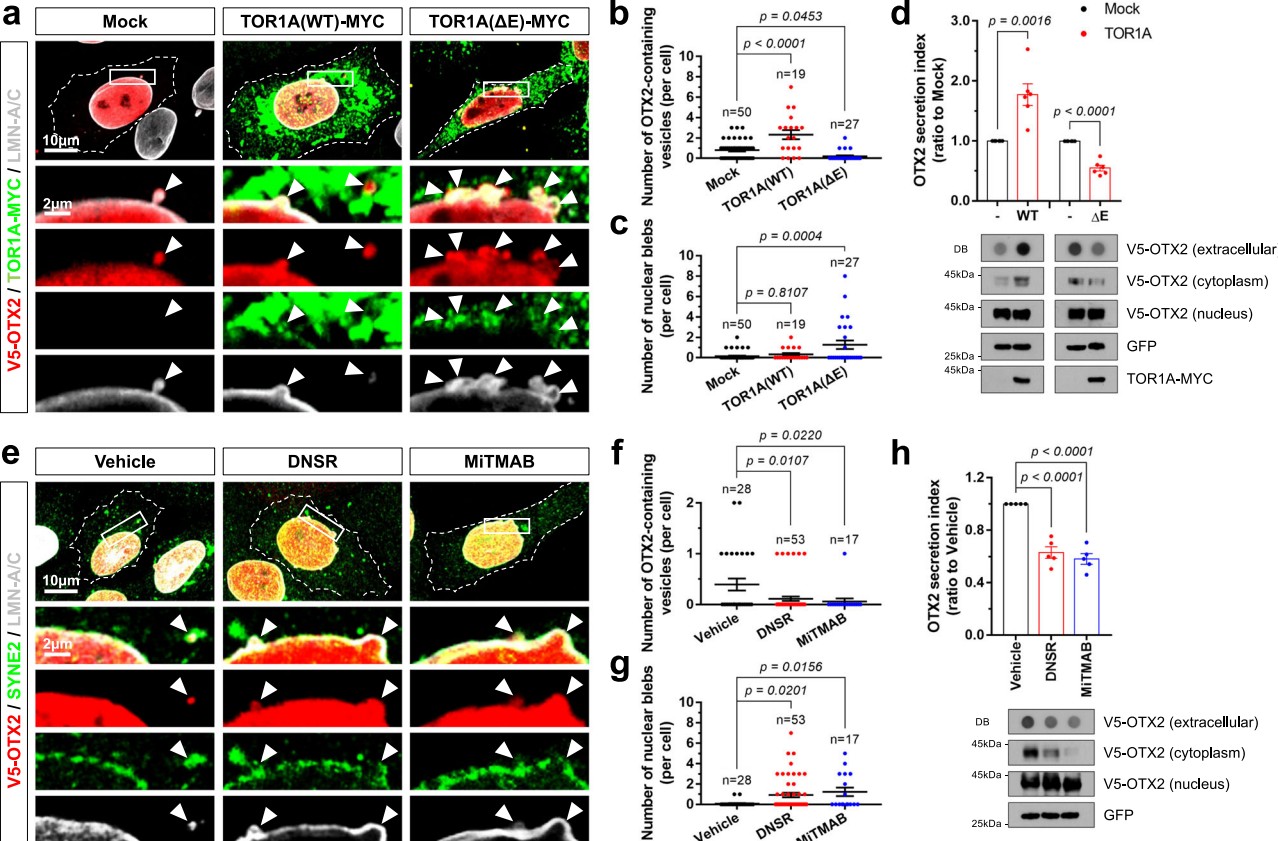

**Fig. 4 | OTX2 is transported to the cytoplasm in nuclear budding vesicles.**
**a** HeLa cells co-expressing V5-OTX2 and MYC-tagged human TOR1A(WT) or TOR1AΔE were stained with anti-V5 and anti-MYC antibodies. Nuclear laminae of the cells were also visualized by immunostaining of LMN-A/C. Arrowheads indicate perinuclear and cytoplasmic V5-OTX2 signals, which are surrounded by signals of TOR1A and LMN-A/C. Dots in the graphs represent the number of OTX2-containing vesicles (**b**) and nuclear blebs (**c**) in a cell, respectively. Long sidebars indicate mean values and error bars denote SEM obtained from 4 independent experiments. The numbers of cells analyzed are shown in the graphs. **d** V5-OTX2 in the growth medium and whole cell lysates of HeLa cells expressing TOR1A(WT)-MYC or TOR1AΔE-MYC was detected by DB and WB, respectively. The cell lysates were fractionated into the nucleus and cytoplasm prior to WB detection. The columns in the graph represent the relative secretion index of OTX2 determined by DB. The values are means and error bars denote with SEM obtained from 6 independent experiments. **e** Chemical inhibitors of DNM, such as DNSR (50 µM) and MiTMAB

(10 µM), were added to the growth medium of OTX2-HeLa cells for 12 h. V5-OTX2 (red), SYNE2 (green), and LMN-A/C (gray) of those cells were then visualized by immunostaining. Arrowheads indicate V5-OTX2 co-localizing with SYNE2 and LMN-A/C in the perinuclear space and cytoplasm. Dots in the graphs represent the number of OTX2-containing vesicles (**f**) and nuclear blebs (**g**) in a cell, respectively. Long sidebars represent mean values, and error bars denote SEM obtained from 4 independent experiments. The numbers of cells analyzed are shown in the graphs. **h** Levels of V5-OTX2 in the growth medium and whole cell lysates of OTX2-HeLa cells treated with DNSR or MiTMAB were detected by DB and WB, respectively. The columns in the graph represent mean values of the relative secretion index of OTX2 determined by DB. Error bars denote SEM obtained from 5 independent experiments. Statistical analysis was performed by one-way ANOVA with Dunnett's multiple comparison test (in **b**, **c**, and **f**–**h**) and two-tailed Student's $t$-test (in **d**). Source data are provided as a Source Data file.

budding vesicles and then transported to the lysosome and extracellular space (Fig. 5a).

## Reduced secretion of Otx2 from the choroid plexus in *Tor1a*$^{\Delta E/+}$ mice

Despite the corresponding increase and decrease of OTX2 nuclear egress by TOR1A(WT) and TOR1AΔE co-expression (Fig. 4a–d), we could not rule out a possibility that the changes resulted from non-specific nuclear membrane dysmorphosis by the overexpressed TOR1A proteins. We, thus, investigated the roles of Tor1a in the unconventional transport of endogenous Otx2 in vivo, by testing whether Otx2 secretion from the ChP is affected in mice expressing the Tor1aΔE mutant[55]. We found that the number and size of nuclear blebs were increased in ChP cells of P30 *Tor1a*$^{\Delta E/+}$ mice (Fig. 5b–d), as seen in cultured HeLa cells expressing TOR1AΔE (Fig. 4a–c) and spinal neurons[55]. Consequently, the amount of Otx2 secreted into the CSF from the ChP cells of *Tor1a*$^{\Delta E/+}$ mice was decreased compared with that of *Tor1a*$^{+/+}$ littermate mice (Fig. 5g, h), although the amount of Otx2 in the ChP was not changed significantly (Fig. 5e, f).

Otx2 secreted from ChP cells was trapped by CSPGs in the perineural net (PNN) (Fig. 3e), which surrounds immature PV neurons in the mouse V1, prior to penetration into the neurons[32,46,47]. This exogenous Otx2, in turn, induces the maturation of PV neurons and further accumulation of CSPG proteoglycans, which can be bound by Wisteria floribunda agglutinin (WFA), in the PNN[47]. In keeping with the decrease in Otx2 secreted into the CSF (Fig. 5g, h), the levels of Otx2 in the V1 were also reduced in *Tor1a*$^{\Delta E/+}$ mice (Fig. 5i, j). Consequently, the numbers of PV- and WFA-positive cells in the V1 of *Tor1a*$^{\Delta E/+}$ mice were decreased compared with those in *Tor1a*$^{+/+}$ littermate mice, whereas the number of calretinin (CR)-positive cortical neurons, which barely exhibit Otx2[56], was not changed (Fig. 5k–m).

The maturation of PV neurons is known to be a critical step in the closure of synaptic plasticity in the V1 for mouse visual maturation[57,58]. In parallel with the decrease in PV neuron numbers in the *Tor1a*$^{\Delta E/+}$ mouse V1 (Fig. 5k, m), the visual acuity of these mice remained significantly lower than that of *Tor1a*$^{+/+}$ littermates (Fig. 5n). These results suggest that the Tor1aΔE mutant exerts negative effects on nuclear

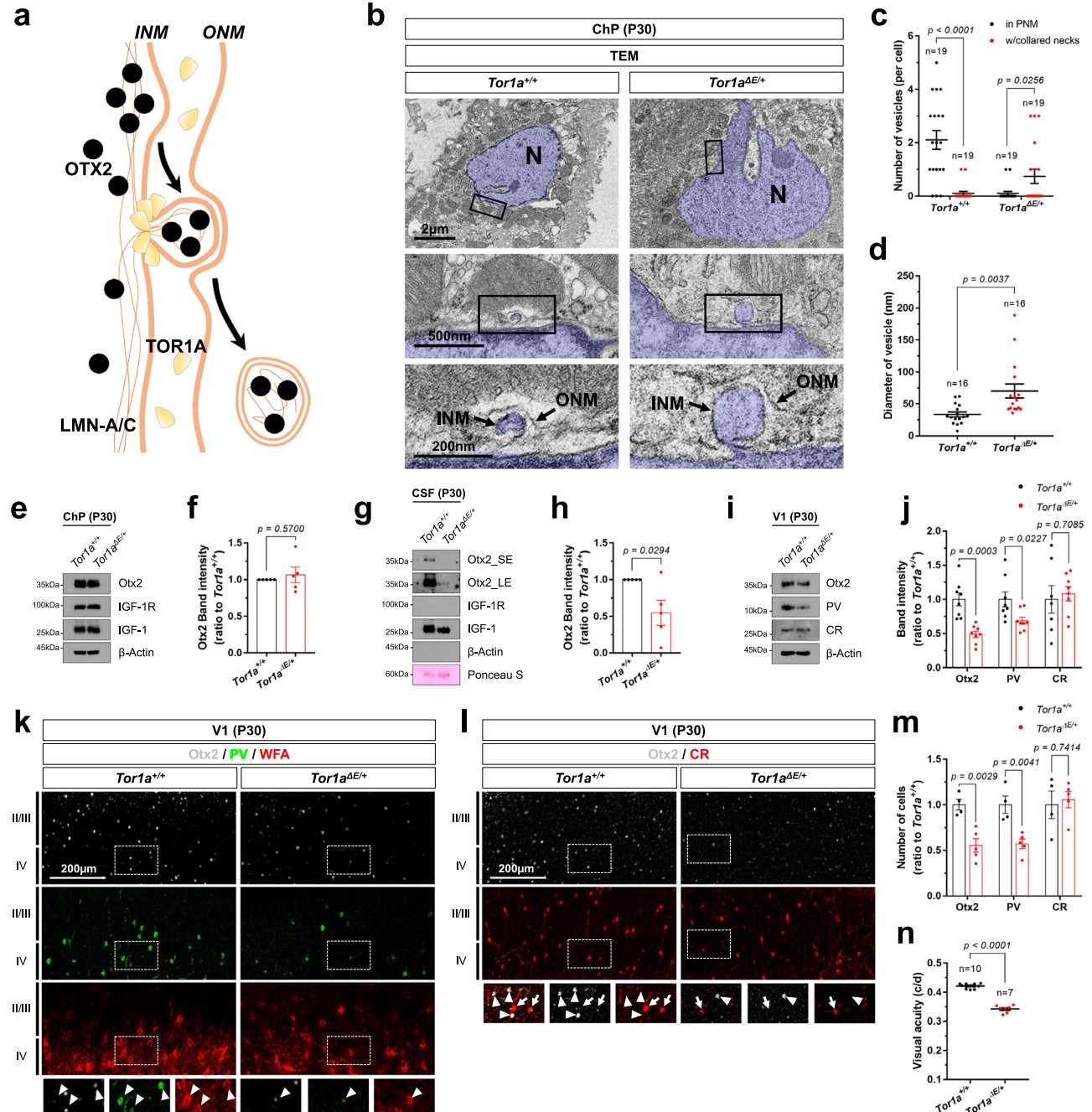

**Fig. 5 | Otx2 secretion is suppressed in *Tor1a^{ΔE/+}* mice. a** The diagram shows the TOR1A-mediated nuclear egress of OTX2, which forms double nuclear membrane vesicles containing LMN-A/C. **b** Sections (70 nm) of ChP from P30 *Tor1a^{+/+}* and *Tor1a^{ΔE/+}* littermate mouse brains were counterstained for electron microscopic observation of intracellular structures. Boxed areas in the top and middle rows are magnified in the middle and bottom rows, respectively. Nuclei (N) of the cells are pseudo-colored in blue. **c** A dot in the graphs represents the number of vesicles in the perinuclear membrane space (PNM) and vesicles with collared necks in a cell. **d** A dot in the graphs represents the average diameters of vesicles in a cell. The numbers of cells analyzed are shown in the graphs. Long sidebars indicate mean values and error bars denote SEM obtained from 8 different samples (4 independent litters). The ChP (**e**), CSF (**g**), and V1 area of cerebral cortex (**i**) were isolated from P30 *Tor1a^{+/+}* and *Tor1a^{ΔE/+}* mice (*n* = 4, each), and the levels of Otx2 in those samples were examined by WB. Relative amounts of proteins loaded in each lane were examined by comparing band intensities of ß-Actin (**e**, **i**) and 60 kDa

protein(s) stained by Ponceau S (**g**). **f**, **h**, **j** The graphs show mean values of WB band intensities. Error bars denote SEM obtained from 5 independent experiments. **k** V1 area of P30 *Tor1a^{+/+}* and *Tor1a^{ΔE/+}* littermate mouse brain sections were stained with anti-Otx2 antibody, anti-PV antibody, and biotinylated WFA. Images in the bottom row are magnified versions of dotted box areas in the top rows. Arrowheads indicate Otx2 signals in PV neurons. **l** Alternatively, the sections were stained with anti-Otx2 and anti-CR antibodies. Arrowheads indicate Otx2-positive cells, and arrows mark CR cells. **m** The graph shows relative numbers of Otx2-, PV-, and CR-positive cells in V1 areas examined by the immunostaining. The error bars denote SEM obtained from 4 different samples (4 independent litters). **n** A dot in the graph represents the average visual acuities of a mouse measured by OptoMotry systems at P30. Long sidebars indicate mean values, and error bars denote SEM. The numbers of mice analyzed from 5 independent litters are shown in the graph. Two-tailed Student's *t*-test was performed for the statistical analysis (in **c**, **d**, **f**, **h**, **j**, **m**, **n**), and source data are provided as a Source Data file.

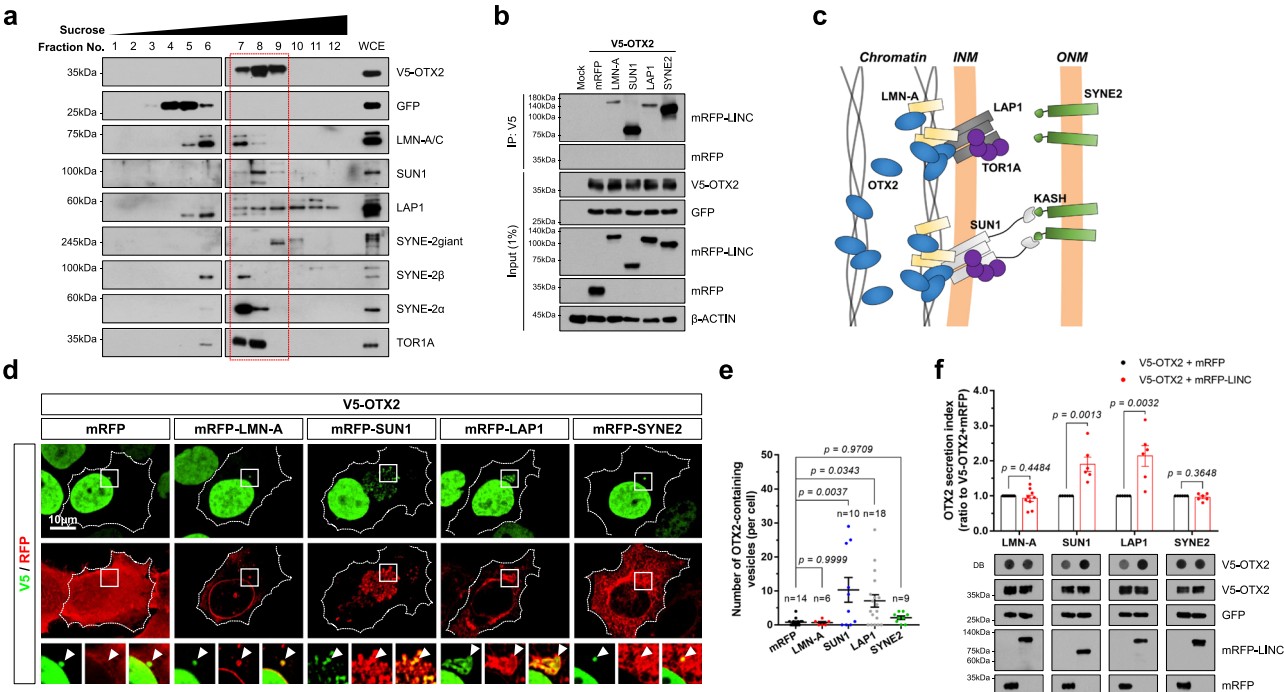

**Fig. 6 | OTX2 binds to the LINC complex. a** Cytoplasm of OTX2-HeLa cells were fractionated by sucrose density gradient. Proteins in 5% of each fraction and 0.2% of WCE were detected by WB. A Dotted red box indicates OTX2-positive fractions. Similar results were obtained in 3 independent experiments, and representative blots are shown. **b** V5-OTX2 was co-transfected into HeLa together with mRFP-tagged LINC components. Interaction between OTX2 and LINC components was examined by co-immunoprecipitation (Co-IP) and WB with indicated antibodies. Similar results were obtained in 3 independent experiments, and representative WB images are shown. **c** Schematic diagram depicts the interaction between OTX2 and the LINC components. **d** HeLa cells overexpressing V5-OTX2 and mRFP-tagged LINC components were stained with anti-V5 and anti-RFP antibodies. Images in the bottom row are magnified versions of dotted box areas in the top rows. Arrowheads point to perinuclear OTX2 puncta. **e** Dots in the graphs represent the number of OTX2-containing vesicles in each cell. Long sidebars indicate mean values and error bars denote SEM obtained from four independent experiments. **f** The levels of extracellular and intracellular V5-OTX2 were examined by DB and WB, respectively, analyses with anti-V5 an antibody. The expression of the LINC components in the cells was also examined by WB using an anti-RFP antibody. The graph shows the mean of the relative secretion index of OTX2. Error bars denote SEM obtained from 6 independent experiments. Statistical analysis was performed by one-way ANOVA with Dunnett's multiple comparison test (in **e**) and two-tailed Student's *t*-test (in **f**). Source data are provided as a Source Data file.

membrane scission and the subsequent intercellular transfer of Otx2 during a critical period of visual maturation in mice.

## OTX2 binds to the LINC complex for nuclear egress

Given the presence of LMN-A/C in OTX2-containing fractions (Fig. 1d, e), we further examined whether nuclear membrane proteins and TOR1A in the internuclear membrane space were also present in OTX2-containing cytoplasmic vesicles. We found that proteins in the inner nuclear membrane, such as LAP1 and SUN1 (Sad1 and UNC84 domain containing 1), and outer nuclear membrane, including SYNE2, were also detected in OTX2-containing cytoplasmic fractions (Fig. 6a). These fractions also contained LMN-A/C and TOR1A. They also partially overlapped with LAMP2 (Fig. 6a), suggesting potential co-transport of OTX2 and these nuclear proteins to the lysosome.

We thus investigated whether these LINC complex components could bind and recruit OTX2 to the nuclear membrane. We used immunoprecipitation to examine the physical interaction between LINC components fused with a monomeric red fluorescent protein (mRFP) and V5-OTX2 in HeLa cells. Our results show that V5-OTX2 co-precipitates SUN1 and SYNE2 more effectively than LMN-A and LAP1 (Fig. 6b). However, given that SYNE2 in the outer nuclear membrane lacks access to OTX2 in the nucleus, SYNE2 might bind indirectly with OTX2 via SUN1 in the inner nuclear membrane (Fig. 6c). Consistent with this, SUN1 and LAP1 co-localized with OTX2 in cytoplasmic puncta (Fig. 4d). Furthermore, the numbers of OTX2 cytoplasmic puncta were increased by co-expression of SUN1 and LAP1, but not LMN-A and SYNE (Fig. 6d, e). LAP1 and SUN1 also significantly enhanced the secretion of OTX2, whereas LMN-A and SYNE2 did not (Fig. 6f). These results

suggest that OTX2 interacts with SUN1 and LAP1 for its entrapment into nuclear budding vesicles that are destined for secretion and/or lysosomal transport.

## LINC is necessary for OTX2 secretion in vivo

We next investigated whether the LINC complexes are necessary for the vesicular trafficking of OTX2. To this end, we fused the KASH2 domain of human SYNE2 to EGFP to generate EGFP-KASH2, which occupied the internuclear membrane space as well as the luminal space of the ER (Fig. 7a–c). EGFP-KASH2 inhibits the interaction of SUN1 in the inner nuclear membrane with the KASH2 domain of SYNE2 in the outer nuclear membrane[59,60], and thus may dissociate the bridge between inner and outer nuclear membranes mediated by the SUN1-SYNE2 complex (Fig. 7a). Consistent with this, the internuclear membrane space occupied by EGFP-KASH2 and surrounded by the SYNE2-positive outer nuclear membrane and LMN-A/C-positive nuclear lamina was expanded significantly compared with the space lacking of EGFP-KASH2 (Fig. 7b, c). However, the number of cytoplasmic OTX2 puncta was decreased in HeLa cells co-expressing EGFP-KASH2, although OTX2-containing perinuclear buds still formed in these cells (Fig. 7b, d). Consequently, cells expressing EGFP-KASH2 exhibited decreased levels of extracellular OTX2 (Fig. 7e, f).

Enlargement of the internuclear membrane space was also observed in vivo. The width of the internuclear membrane space was significantly increased in ChP cells expressing *EGFP-KASH2* after excision of the *loxP-STOP-loxP* (LSL) cassette by tamoxifen-activated CreER recombinase in *LSL-KASH2;FoxJ1-CreER* mice[60,61] compared with that in *FoxJ1-CreER* littermate mouse ChP cells (Figs. 8a, b and 3a and

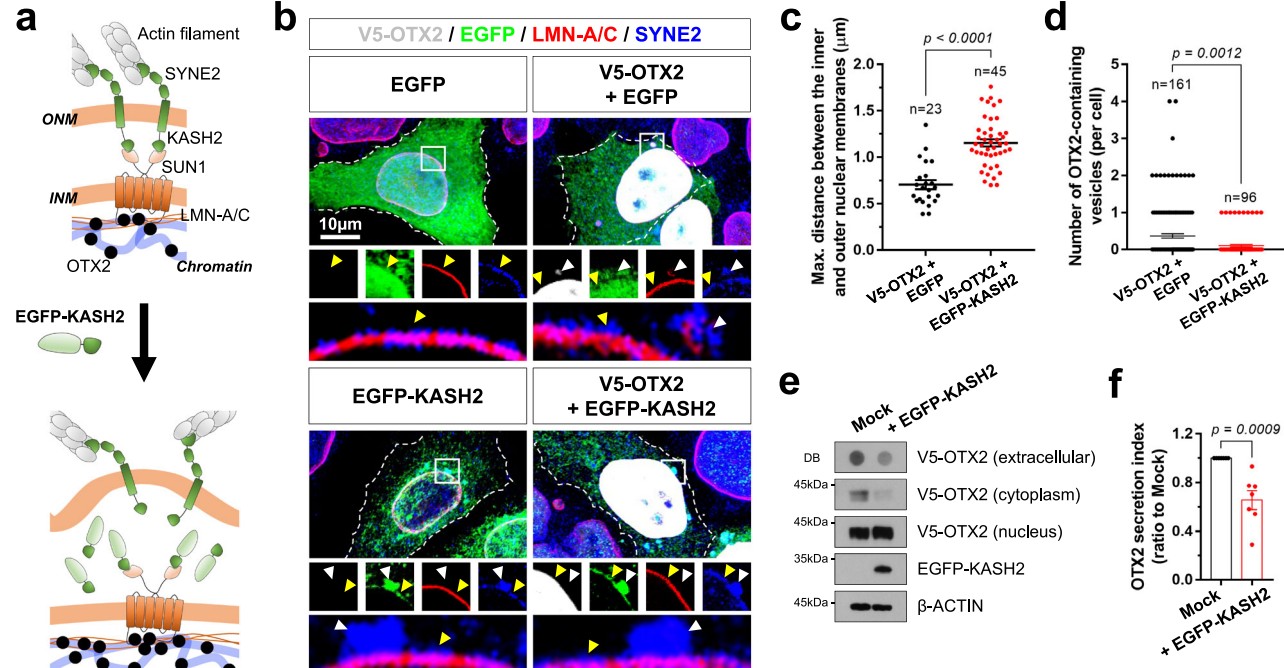

**Fig. 7 | OTX2 secretion is suppressed upon disruption of the LINC complex.**
**a** Schematic diagram that shows the disruption of LINC complex by EGFP-KASH2.
**b** HeLa cells overexpressing V5-OTX2 together with EGFP or EGFP-KASH2 were stained with antibodies detecting V5, EGFP, LMN-A/C, and SYNE2. White and yellow arrowheads indicate flat and bulged areas of the nuclear envelope, respectively. A dot in the graphs represents the maximum (Max.) distance between the inner and outer nuclear membranes (**c**) and the number of OTX2-containing perinuclear and cytoplasmic vesicles (**d**) in a cell. Long sidebars indicate mean values and error bars denote SEM obtained from 6 independent experiments. The numbers of cells

analyzed are shown in the graph. **e** V5-OTX2 in the growth medium and whole cell lysates of HeLa cells expressing Mock or EGFP-KASH2 was detected by DB and WB, respectively. The cell lysates were fractionated into the nucleus and cytoplasm prior to WB detection. Co-expressed Mock and EGFP-KASH2 in the cells was also determined by WB. **f** The graph shows the mean relative secretion index of OTX2. Error bars denote SEM obtained from 7 independent experiments. Two-tailed Student's *t*-test was performed for the statistical analysis (in **c**, **d**, **f**), and source data are provided as a Source Data file.

Supplementary Fig. 8). This genetic manipulation also caused an increase in the number of single-layer vesicles in the internuclear space, but decreased the number of double-layered nuclear budding vesicles (Fig. 8c).

These alterations in nuclear membrane structures were accompanied by a decrease in secretion of Otx2 into the CSF of *LSL-KASH2;FoxJ1-CreER* mice compared with that of *FoxJ1-CreER* littermate mice (Fig. 8f, g), but without significantly changing Otx2 level in the mouse ChP cells (Fig. 8d, e). These results suggest that the LINC-mediated interaction between inner and outer nuclear membranes is necessary for the secretion of Otx2 from the ChP cells. Reduced transfer of Otx2 and a consequent decrease in the number of PV neurons were also observed in the mouse V1 (Fig. 8h–k), changes that led to a decrease in the visual acuity of *LSL-KASH2;FoxJ1-CreER* mice (Fig. 8l).

**Nuclear OTX2 forms aggregates upon failure of vesicular egress**
Despite the decrease in cytoplasmic and extracellular Otx2 in cells expressing Tor1AΔE and EGFP-KASH2 (Figs. 4a, b, d, 5g, h, 7e, f and 8f, g), we found the levels of Otx2 in mouse ChP cells expressing Tor1AΔE and EGFP-KASH2 were not changed significantly in comparison with those in cells expressing WT TOR1A (Figs. 5e, f and 8d, e). OTX2 tended to form insoluble protein aggregates in vitro and in bacterial cells[62], and could also be detected in the detergent-insoluble fraction of HeLa cell lysates (Fig. 1c and Supplementary Fig. 1c, d). These observations suggest that the intrinsic instability of OTX2 might trigger vesicle-mediated nuclear egress to prevent it from forming of insoluble protein aggregates in the nucleus.

We thus examined whether the expression of TOR1AΔE can result in the formation of OTX2 aggregates in the nucleus. We found

the levels of OTX2 in detergent-insoluble fraction were elevated in TOR1AΔE-expressing cells compared with those in WT TOR1A-expressing cells in culture and in vivo, although OTX2 in the soluble fractions was not significantly changed (Fig. 9a–d). Furthermore, active caspase-3 (Casp3)-positive apoptotic cells were detected sparsely in the ChP of *Tor1a^{ΔE/+}*, but were undetectable in the ChP of their *Tor1a^{+/+}* littermates (Fig. 9e, f). These results suggest moderate ongoing degeneration of ChP cells in *Tor1a^{ΔE/+}* mice, although it is not clear whether the death of ChP cells was related to insoluble OTX2.

Collectively, our results suggest that OTX2 is exported from the nucleus via an unconventional vesicular transport process. This process encompasses a series of events that begins with the binding of OTX2 to SUN1, which forms a LINC complex with SYNE, and LAP1, which binds TOR1A in the internuclear membrane space (leftmost diagram in Fig. 9g). OTX2 molecules, which can associate each other[63], then cluster the LINC complex to form a double-layered nuclear membrane bud. LAP1 in the bud recruits TOR1A to the base of the inner nuclear membrane bud, allowing TOR1A to cleave the inner nuclear membrane (second leftmost diagram in Fig. 9g). This gives rise to a vesicle with a single membrane layer in the internuclear membrane space. However, given the connection of inner and outer nuclear membranes by the LINC complex, a double-layered nuclear membrane vesicle containing OTX2 would form preferentially through the coordinated actions of TOR1A and DNM, which might be involved in the scission of the outer nuclear membrane (second rightmost diagram in Fig. 9g). However, OTX2 would be accumulated to form insoluble aggregates in the nucleus upon the failure of the inner nuclear membrane scission, such in the cells expressing TOR1AΔE (rightmost diagram in Fig. 9g).

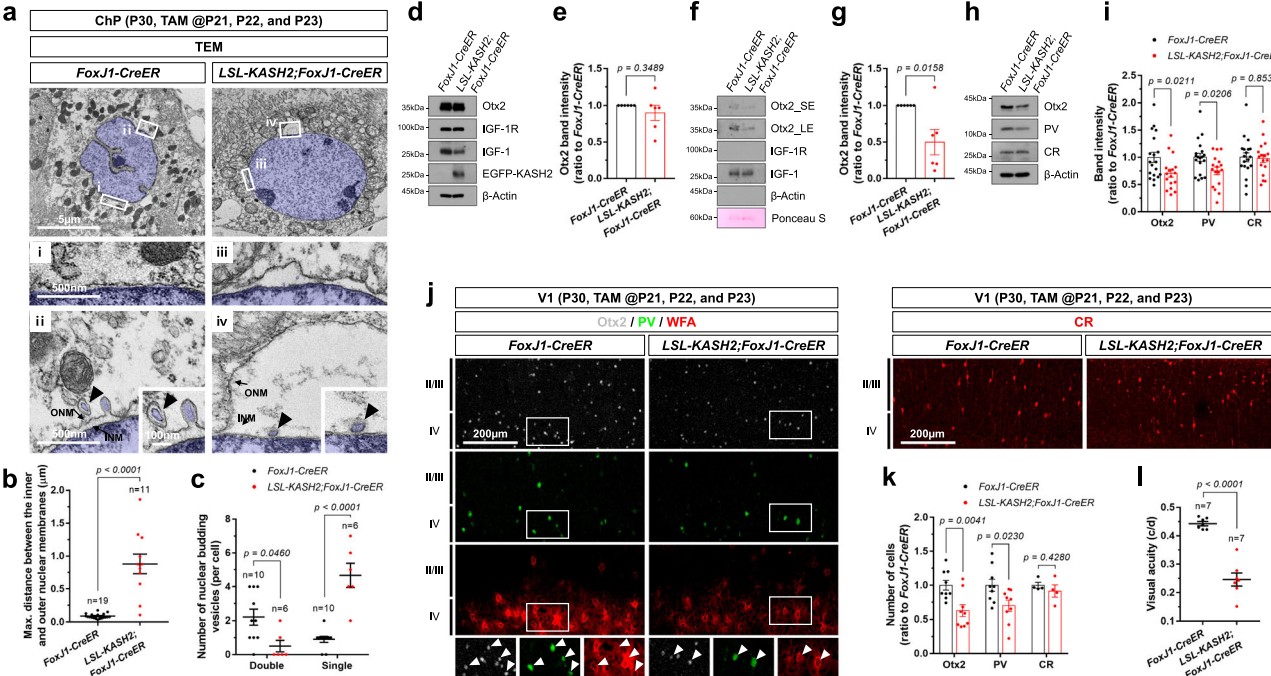

**Fig. 8 | Otx2 secretion is suppressed *LSL-KASH2;FoxJ1-CreER* mice. a** Sections (70 nm) of ChP from P30 *FoxJ1-CreER* and *LSL-KASH2;FoxJ1-CreER* mouse brain were counterstained for ultrastructural imaging by electron microscope. The nuclei of the ChP cells are pseudo-colored in blue. In the mice, CreER recombinase was activated by injection of tamoxifen at P21, P22, and P23. (i - iv) Magnified versions of the corresponding boxed areas on the top row. Arrowheads indicate the nuclear membrane budding vesicles. A dot in the graphs represents Max. distance between the inner and outer nuclear membranes (**b**) and the number of OTX2-containing perinuclear and cytoplasmic vesicles (**c**) in a cell. Long sidebars indicate mean values, and error bars denote SEM obtained from 8 different samples (4 independent litters). The numbers of cells analyzed are shown in the graphs. The ChP (**d**), CSF (**f**), and V1 area of the cerebral cortex (**h**) were isolated from P30 *FoxJ1-CreER* and *LSL-KASH2;FoxJ1-CreER* mice (*n* = 4, each), and the levels of Otx2 in those samples were examined by WB. Relative amounts of proteins loaded in each lane were determined by comparing band intensities of ß-Actin (**d, h**) and 60 kDa protein(s) stained by Ponceau S (**f**). **e, g, i** The graphs show the mean values of WB band intensities. Error bars denote SEM obtained from 5 independent experiments. **j** V1 area of P30 *FoxJ1-CreER* and *LSL-KASH2;FoxJ1-CreER* littermate mouse brain sections were stained with anti-Otx2 antibody, anti-PV antibody, and biotinylated WFA. Images in the bottom row are magnified versions of dotted box areas in the top rows. Arrowheads indicate Otx2 signals in PV neurons. Alternatively, the sections were stained with anti-CR antibody. **k** The graph shows relative numbers of Otx2-, PV-, and CR-positive cells in V1 areas examined by the immunostaining. The error bars denote SEM obtained from 4 different samples (4 independent litters) **l**, A dot in the graph represents the average visual acuities of a mouse measured by OptoMotry systems at P30. Long sidebars indicate mean values and error bars denote SEM. The numbers of mice analyzed from 5 independent litters are shown in the graph. Statistical analysis (in **b, c, e, g, i, k, l**) was performed by Student's *t*-test (two-tailed), and source data are provided as a Source Data file.

The resultant double-layered vesicle is then transported to the plasma membrane and intracellular organelles, mostly the lysosome (Figs. 1f and 2d), through SYNE-associated cytoskeletons. The outer layer of the vesicle then fuses to plasma and lysosomal membranes to release the single-layered vesicle into the extracellular space and lysosome, respectively (Fig. 2a–d, g, h). The OTX2-containing inner vesicular membrane is shed by lipases to expose OTX2 in the extracellular space for intercellular transfer and in the lysosome for degradation (Supplementary Fig. 3). Lysosomes containing OTX2 could also fuse to the plasma membrane to release OTX2. Therefore, defects in the steps of this unconventional OTX2 nuclear export would abrogate intercellular transfer of OTX2 from ChP cells to PV neurons, which is necessary for the maturation of visual neural circuits (Figs. 5n and 8h).

## Discussion

Cellular OTX2 levels were increased by lysosomal dysfunction but not by inhibition of proteasomal degradation—the most common decay mechanism for cytoplasmic and nuclear proteins[15] (Fig. 2 and Supplementary Fig. 2). These results suggest that lysosomal transport is the primary cellular mechanism for OTX2 degradation. This unconventional protein decay mechanism could also apply to other nuclear proteins, which are resistant to proteasomal degradation. However, additional studies are needed to confirm that this is a common feature of sHPs.

Lysosomal dysfunction increased both of intracellular and extracellular OTX2 levels (Fig. 2a–c). This increase in extracellular OTX2 could be interpreted as reflecting one of two possible mechanisms. First, OTX2 is transported to the lysosome for degradation and then OTX2 survived in the lysosome is released into the extracellular space through fusion of lysosomal membrane to the plasma membrane. In support of this, localization of OTX2 in LAMP2-positive lysosomes became more evident in cells treated with BafA1 or CQ (Fig. 2d, e). Alternatively, OTX2 could be secreted without the mediation of lysosomes. Treatment with BafA1 or CQ should cause an accumulation of proteins and lipids that exceeds the capacity of lysosomes, suppressing the merger of additional vesicles with the lysosome. Under these lysosome-malfunctioning conditions, vesicles carrying OTX2 might be transported to an alternative target—the plasma membrane—and release OTX2 into the extracellular space. Therefore, real-time tracking of OTX2 in BafA1-treated cells is necessary to determine whether vesicles detour to the plasma membrane under conditions of lysosomal dysfunction.

Given the sensitivity of OTX2 to lysosomal lipase inhibitors (Supplementary Fig. 4), OTX2 is likely present in the lysosomal luminal vesicles. Therefore, OTX2 in the cytoplasm should be transported in the multi-layered membrane vesicles that fuse to the lysosome and release their inner vesicles into the lysosome. MVBs and autophagosomes capture cytoplasmic proteins and wrap them into their luminal vesicles before fusing to the lysosome[38]. However, as OTX2 was not

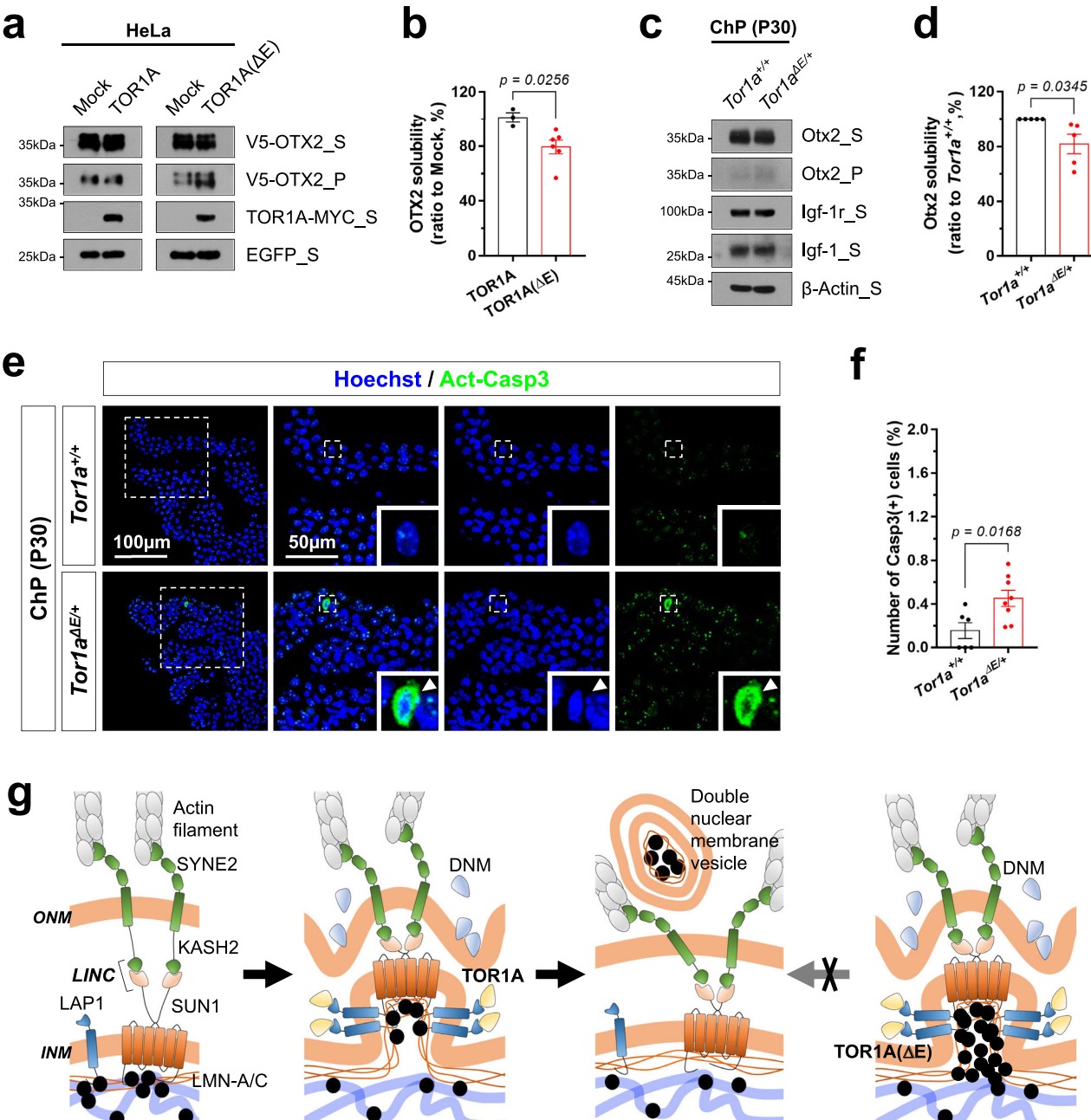

**Fig. 9 | Excessive OTX2 forms insoluble aggregates in the nucleus. a** HeLa cells overexpressing V5-OTX2 together with TOR1A-MYC or TOR1AΔE-MYC were lysed in RIPA buffer. The soluble supernatant fractions (S) of the cell lysates were separated from insoluble precipitate fractions (P) by centrifugation. Relative level of V5-OTX2 in each fraction were then examined by WB with anti-V5 antibody. The levels of EGFP, which is co-transcribed with V5-OTX2 in a same mRNA, and MYC-tagged TOR1A and TOR1AΔE was also investigated by WB. **b** OTX2 solubility was determined by dividing band intensity of V5-OTX2_S by that of total (V5-OTX2_S + V5-OTX2_P). Relative solubility values to Mock (pCAGIG)-transfected samples were obtained and shown in the graph. The graph shows mean OTX2 solubility values. Error bars denote SEM obtained from 4 independent experiments. **c** ChP cells were isolated from P30 *Tor1a*⁺/⁺ and *Tor1a*ΔE/⁺ mice (*n* = 4, each) and S and P fractions were separated from the cell lysates. Otx2 in each fraction were then detected by WB with anti-Otx2 antibody. The levels of Igf-1, which is secreted by the ChP cells, and

Igf-1r, which is expressed in ChP cell membrane, were also determined by WB. Relative amounts of proteins loaded in each lane were determined by comparing WB band intensities of ß-Actin. **d** The graph shows relative Otx2 solubility. The values are the means and error bars denote SEM obtained from 5 independent experiments. **e** Brain sections of P30 *Tor1a*⁺/⁺ and *Tor1a*ΔE/⁺ littermate mice were stained with an antibody detecting activated caspase-3 (Act-Casp3). Nuclei of the cells in the sections were visualized by staining with Hoechst. The images show the ChP areas of lateral ventricle. Three right images are magnified versions of dotted box areas in the leftmost column. Insets are magnified images of the areas pointed by arrowheads. **f** The graph shows mean numbers of Act-Casp3-positive cells in the sections. Error bars denote SEM obtained from 6 samples (3 independent litters). **g** Schematic diagram depicts the budding vesicle-mediated nuclear egress of OTX2. Two-tailed Student's t-test was performed for the statistical analysis (in **b**, **d**, **f**), and source data are provided as a Source Data file.

detected in MVBs and autophagosomes (Fig. 1f, g), it seems unlikely that naked OTX2 in the cytoplasm is transported to the lysosome via MBVs or autophagosomes. Instead, OTX2 was detected in the nuclear membrane buds (Fig. 3a, f), which can develop to vesicles through

membrane severing by TOR1A. This vesicular trafficking is used for the nuclear egress of HSV and RNP[10–12]. To form nuclear budding vesicles, HSV uses pUL31 to bind pUL34, which integrates to the nuclear inner membrane and disrupts the nuclear lamina and LINC complex[11,12].

Given the co-precipitation of OTX2 with SYNE2, which resides in the outer nuclear membrane (Fig. 6b, c), the double-layered nuclear membrane structure supported by the LINC complex is likely maintained to form double-layer membrane vesicles containing OTX2 (Fig. 3a, c, f, h). The outer nuclear membrane of these double-layered vesicles might fuse to the lysosomal membrane to release the inner nuclear membrane vesicles containing OTX2 in the lysosome.

Although the Otx2 levels were significantly decreased in the CSF of *Tor1a^{ΔE/+}* and *LSL-EGFP-KASH2;FoxJ1-CreER* mice, those in their ChP cells were not significantly altered (Figs. 5e–h and 8d–g). Only a small portion (<5%) of cellular HPs can be secreted[30]. Therefore, Otx2 molecules that fail to exit the nucleus in mouse ChP cells might not be sufficient to significantly increase the total amount of Otx2. The extranuclear Otx2, however, could form aggregates, as overexpressed OTX2 was found to be accumulated and form insoluble aggregates in the nucleus upon suppression of vesicular egress by TOR1AΔE (Fig. 9a–d). Thus, the unconventional nuclear export might eliminate OTX2 from the nucleus before it forms aggregates.

RR-to-AA mutations in the NLS not only inhibited translocation of OTX2 to the nucleus, they also decreased its solubility in the cytoplasm (Supplementary Fig. 1d, e), suggesting that these RR residues are critical not only for nuclear transport but also for stabilization of the protein. OTX2 forms an insoluble aggregate upon expression in bacteria[62], suggesting that chaperones present in animal cells, but absent in bacterial cells, might support OTX2 to form a stable structure. Thus, importin, which is a co-translational chaperone for nuclear proteins[64], might bind the NLS and solubilize OTX2 before it transfers OTX2 to its binding partners in the nucleus. One of the most prominent nuclear binding partners that might stabilize OTX2 is DNA, since OTX2 remains soluble in the presence of target DNA[63]. However, given the dynamic association and dissociation of transcription factors with their target DNA[65], OTX2 would be released from the DNA and revert to an unstable structure unless it found another binding partner in the nucleus. Given the detergent solubility of OTX2 in the cytoplasmic vesicles budded from the nucleus (Figs. 1c, 3a, f and Supplementary Fig. 1c), the LINC complex might also stabilize OTX2 in the nucleus and vesicles.

Another sHP, EN2, was suggested to bind phosphoinositide 4,5-bisphosphate (PIP$_2$) in the inner leaflet of the plasma membrane for secretion[66]. It was also found in the caveolae[67]. These reports suggest that EN2 may be directly associated with the nuclear membrane. We observed that EN2 forms cytoplasmic puncta in HeLa cells (Fig. 1a), prompting us to speculate that EN2 binds to PIP$_2$ in the inner nuclear membrane prior to budding. This would contrast with the binding of OTX2 to SUN1 and LAP1 proteins in the inner nuclear membrane for vesicular entrapment. These observations suggest that sHPs could use multiple paths to bind the inner nuclear membrane for vesicular nuclear egress.

Nuclear bud-mediated decay mechanisms have also been identified in *Saccharomyces cerevisiae*. In a process termed piecemeal microautophagy of the nucleus (PMN), formation of a nucleus-vacuole (NV) junction initiates nuclear membrane budding for subsequent fission and ingestion of nuclear vesicles by the vacuole[68]. NV junction formation is mediated by specific protein–protein interactions between the nuclear membrane protein, nucleus-vacuole junction protein 1 (Nvj1p), and the vacuolar membrane protein, Vac8p[69]. However, no *Nvj1p* or *Vac8p* orthologs have been identified in animal cells. Functional homologs of PMN components for transporting nuclear budding vesicles to the lysosome, an organelle homologous to the yeast vacuole, could be present in animal cells[70]. Given that OTX2 binds to the LINC complex for the transport of nuclear budding vesicles to the lysosome, our data suggest that the functions of Nvj1, which spans two nuclear membranes, could be replaced by SYNE in the outer nuclear membrane and SUN1 in the inner nuclear membrane. However, functional ortholog(s) for yeast Vac8 remain unknown.

Conversely, TOR1 orthologs are not found in the yeast genome, although scission of nuclear membranes is required for the formation of intra-vacuolar vesicles. This suggests that functional ortholog(s) of TOR1 might be present in the internuclear membrane space and ER lumen. Several AAA + ATPases, including p97, which is involved in ER membrane fusion[71,72], have been identified in the ER. These are also conserved in yeast[73], suggesting potential roles for those ATPases in scission of the inner nuclear membrane during PMN. Alternatively, the physical force derived from Nvj1-Vac8 clustering might result in scission. Clearly, future studies will be required to elucidate the scission mechanism of PMN.

In contrast to direct ingestion of nuclear vesicles by the vacuole, the mammalian lysosome is unlikely to ingest nuclear budding vesicles, given that their size (~100–500 nm[74]) is not much larger than the nuclear budding vesicles (>250 nm[14]). Thus, nuclear budding vesicles might be transported to the lysosome via the cytoskeleton through recognition by SYNE-associated motor proteins. This suggests that vesicular nuclear-lysosome trafficking might have coevolved with the lysosome in animal cells.

## Methods

### Experimental model and subject details

**Animals.** *Tor1a^{ΔE/+}* (*Tor1a^{tm2Wtd}/J*) mice were purchased from Jackson Laboratory. *LSL-KASH2* (*Tg(CAG-LacZ/EGFP-KASH2)*) and *FoxJ1-CreER* (*Tg(FOXJ1-cre/ERT2)1Blh*) mice were generated as described in the previous reports[60,75], and were bred to generate *LSL-KASH2;FoxJ1-CreER* mice. The mice were maintained in the specific pathogen-free facility of KAIST Laboratory Animal Resource Center. All of the animals were handled according to approved institutional animal care and use committee (IACUC) protocols (#2012-37) of Korea Advanced Institute of Science and Technology (KAIST).

**Cell culture and establishment of OTX2-HeLa cell-line.** HeLa cells were incubated in Dulbecco's Modified Eagle Medium (DMEM; HyClone, SH30243) supplemented with 10% fetal bovine serum (FBS; Gibco, 12484010) and 1% penicillin/streptomycin (P/S; Gibco, 15140-122) and maintained in a humidified chamber with 5% CO$_2$ at 37 °C. To establish HeLa cells expressing OTX2 constitutively, V5-OTX2-IRES-EGFP DNA fragments were cloned into pCAG plasmid vector, and then neomycin resistance gene (Neo$^R$) was introduced to the vector generate pCAG-V5-OTX2-IRES-EGFP-Neo$^R$ construct. This DNA construct was linearized for the transfection into HeLa cells by GenJet™ Plus Transfection Reagent (SignaGen Laboratories, SL100499). Growth medium of the cells was then replaced by DMEM containing 1 mg/mL geneticin (G418; AG Scientific, G-1033), 10% FBS, and 1% P/S at 16 h post transfection. The cells were maintained in this selection medium, which was changed every 2 days until the cells formed colonies. Cells in the colonies were dispersed by treating with TrypLE™ Express Enzyme (Invitrogen, 12604013) and then collected by centrifugation at 500rcf for 10 min. The cells were resuspended in Hanks' Balanced Salt Solution (HBSS; Thermo Scientific, 88284) supplied with 2% FBS prior to the isolation of individual cells expressing GFP by a BD FACSAria II (BD). The FACS-isolated OTX2-HeLa cells were further cultured in the selection medium. These cells were treated with pharmacological inhibitors listed in Supplementary Table 1 prior to investigating the effects of those reagents on OTX2 trafficking.

### Experimental methods

**Subcellular fractionation and sucrose density gradient centrifugation.** OTX2-HeLa cells ($2 \times 10^7$) were scrapped from the culture plates and resuspended in phosphate-buffered saline (PBS; Gibco, 10010023) to separate the nucleus from cytoplasmic fraction using NE-PER™ Nuclear and Cytoplasmic Extraction Reagents (Thermo Scientific, 78833). Cytoplasmic fractions were then centrifuged at 500rcf for 10 min at 4 °C to collect the supernatant (S1). Next, the S1 fractions

were centrifuged at 2000 rcf for 20 min at 4 °C. The resultant S2 samples were carefully loaded on top of sucrose solution forming the gradients (5–45%), which were generated by Linear Gradient Makers (C.B.S. Scientific, GM-20) according to manufacturer's instructions, prior to ultracentrifugation at 257,000rcf for 18 h at 4 °C. After ultracentrifugation, each fraction was collected in an equal volume using the Fraction Recovery System (Beckman Coulter, 270-331580) from bottom to top.

**Dot-blot analysis.** Relative amounts of secreted OTX2 in the growth medium were determined as described previously[30]. Briefly, growth medium of HeLa cells was replaced by FreeStyle™ 293 Expression Medium (Gibco, 12338018) containing heparin (10 mg/ml; Millipore, 375095) for 3 h. The heparin-containing medium was then collected and loaded into the Minifold® I 96 well dot-blot (DB) array system (Whatman, 10447900) to transfer proteins in the medium to poly-vinylidene fluoride (PVDF; Millipore, IPVH00010) membrane, according to the manufacturer's instructions, while proteins in the cell lysates were analyzed by western blot (WB). The DB and WB membranes were probed with primary antibodies prior to detection of those antibodies by horseradish peroxidase (HRP)-conjugated secondary antibodies. Chemiluminescence of intensities of the dots and WB bands on the blots were measured using ImageJ software (NIH). Relative secretion index was calculated as described previously[30].

**Immunostaining.** Cells on the cover glasses were fixed in 4% paraformaldehyde (PFA; Sigma-Aldrich, 158127)/PBS for 20 min and then permeabilized in 0.2% Tween-20 (Sigma-Aldrich, P9416)/PBS for 10 min. Next, cells were incubated in a blocking solution (0.2% Triton X-100, 10% normal donkey serum (Jackson ImmunoResearch, 071-000-121), 2% bovine serum albumin (BSA; Sigma-Aldrich, A7906), and 1 M glycine (Sigma-Aldrich, G8898)) in PBS for 1 h at room temperature. Cells were subsequently incubated with primary antibody diluted in antibody dilution buffer (Agilent Technologies, S3022) for 16 h at 4 °C. Afterward, the cells were further incubated in appropriate secondary antibodies and mounted with a fluorescence mounting medium (Agilent Technologies, S3023). Immunofluorescence signals were analyzed using a confocal microscope (Olympus, FV3000). For quantitative analyses, cells positive for immunostaining signals in each sample were counted in a blinded fashion, and the samples were matched after counting.

For immunohistochemistry (IHC), mouse brain tissues, fixed by cardiac perfusion of 4% PFA/PBS, were isolated and incubated in the same solution for 16 h at 4 °C. The tissues were then transferred to 30% sucrose/PBS and put at 4 °C until the tissues sedimented completely. Tissues were embedded in an O.C.T. compound cryostat embedding medium (Scigen, 23-730-625) and frozen on dry ice. Sections were obtained in 15 μm thickness for the ChP and 30 μm thickness for the cerebral cortex by Cryostat (Leica Biosystems, CM1860). The sections were permeabilized in 0.2% Triton X-100 (Sigma-Aldrich, X100)/PBS for 30 min and optionally incubated in 10 mM citric acid (Sigma-Aldrich, 251275)/PBS for 1 h at 60 °C. The following process is the same as the immunostaining procedure described above. The list of antibodies used in this study is provided in Supplementary Table 2.

**Electron microscopy.** For transmission electron microscopy (TEM) analysis, perfusion-fixed mouse brain tissues were additional fixed in 2% PFA (Electron Microscopy Sciences, 15714), 2% glutaraldehyde (GA; Electron Microscopy Sciences, 16220)/PBS for 2 h at 4 °C, and then post-fixed with 2% osmium tetroxide (OsO₄; Electron Microscopy Sciences, 19140) for 1 h on ice. Following a series of ethanol dehydration, samples were stained with 0.5% uranyl acetate (UA; Electron Microscopy Sciences, 22400)/DDW for 2 h, and embedded in EMbed 812 resin (Electron Microscopy Sciences, 14120). Ultrathin sections (70 nm) were prepared using an ultramicrotome (Leica Microsystems,

Ultracut EM UC7), and then counterstained with UA and lead citrate (Electron Microscopy Sciences, 22410).

For cryo-immunogold staining, the cultured HeLa cells and tissues were fixed in 4% PFA, 0.05% GA/PBS for 20 min at 4 °C and incubated in 2.3 M sucrose (Sigma-Aldrich, S0389)/PBS for 24 h at 4 °C. The cells and tissues were frozen and stored in liquid nitrogen. Ultrathin sections (50 nm), collected on 100–150 mesh copper grids, were incubated in a 50 mM glycine/PBS for 10 min and blocked with 5% BSA, 0.1% cold fish gelatin (Electron Microscopy Sciences, 25560)/ PBS for 10 min. Sections were subsequently incubated in 0.1% BSA-c™ (Electron Microscopy Sciences, 25557)/PBS containing primary antibody for 30 min at room temperature. And then, sections were incubated in gold nanoparticles-conjugated secondary antibody for 30 min prior to staining with 2% uranyl acetate (UA) for 5 min. The grids were coated with 0.2% UA, 2% methylcellulose (Sigma-Aldrich, M7140)/DDW and kept in a constant temperature and humidity chamber.

For RT-immunogold staining, perfusion-fixed tissues were incubated in 4% PFA, 0.05% GA/PBS for 15 min and then in 4% PFA/PBS for 30 min at RT. The tissues were rinsed with 0.1% BSA-c™/PBS and blocked with 0.1% saponin (Sigma-Aldrich, SAE0073), 5% BSA, 50 mM ammonium chloride (Sigma-Aldrich, 254134)/PBS for 30 min. Samples were subsequently incubated in blocking solution containing primary antibody for 24 h at 4 °C. Then, samples were incubated in gold nanoparticles-conjugated secondary antibody for 1 h at RT, and gold nanoparticles were visualized using silver enhancement kit (Electron Microscopy Sciences, 25521) according to the manufacturer's instructions. Gold-labeled samples were incubated with 2% OsO₄ and 1.5% potassium ferrocyanide/0.1 M sodium cacodylate buffer (pH 7.4) for 1 h on ice. Following a series of staining preocedure in a solution containing 1% thiocarbohydrazide (Electron Microscopy Sciences, 21900), 2% OsO₄, and 1% UA, samples were dehydrated using a gradient ethanol series and embedded in EMbed 812 resin. Ultrathin sections (70 nm) were counterstained with UA and lead citrate to visualize electron-dense areas. The immunostained sections on grid were then examined using the Bio-HVEM System (JEOL, JEM-1400 Plus) operating at 120 kV.

**Mouse visual acuity test.** Mouse visual acuity test was done using the OptoMotry HD System (CerebralMechanics) as described previously[76]. Mice were placed in a testing arena, which is a virtual cylinder surrounded by a quadrangle of computer monitors projecting a vertical sine wave grating pattern. The OptoMotry software randomly projects grating patterns onto the monitors, and the mouse head rotates slowly following the grating patterns. When the mouse stopped tracking, the spatial frequency thresholds were measured with OptoMotry software according to manufacturer's instructions.

### Statistics and reproducibility
**Statistical analysis.** GraphPad Prism (GraphPad Software, v7.00) software was used to analyze data in this study. All data were displayed as the mean and standard error of the mean (SEM). In addition, two-tailed Student's $t$-test and one-way ANOVA with Dunnett's multiple comparison test were performed to confirm the significant difference between the data. A $p$-value below 0.0500 was then considered statistically significant.

### Reporting summary
Further information on research design is available in the Nature Portfolio Reporting Summary linked to this article.

## Data availability
All data generated or analyzed during this study are included in this published article and Supplementary Information. Source Data are also provided with this paper. Source data are provided with this paper.

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

## Acknowledgements

We appreciate Dr. Brigit Hogan for a generous gift of *FoxJ1-CreER* mice. We also thank Dr. Xandra Breakfield for providing us with the DNA construct of TOR1AΔE. This work is supported by National Research Foundation of Korea (NRF) grants (NRF-2022R1A2C3003589; NRF-2018R1A5A1024261) funded by Korean Ministry of Science and ICT (MSIT) and the International Collaboration Initiative grant (KAIST-N11210255) funded by KAIST, South Korea.

## Author contributions

J.W.P. and J.W.K. wrote the manuscript; J.W.P., E.J.L., E.M., H.L.K., and N.S.K. designed and performed the experiments and analyzed the data; D.H. provided an experimental model and platform; I.B.K., H.S.K., and J.W.K. supervised the project.

## Competing interests

The authors declare no competing interests.
