## [Peer Review File · Nature Communications]

Orthodenticle homeobox 2 is transported to lysosomes by nuclear budding vesiclesREVIEWER COMMENTS

Reviewer #1 (Remarks to the Author):

Although homeoprotein (HP) transfer is a now well-established signaling pathway, the unconventional mechanisms for HP internalization and secretion are yet to be fully understood. The manuscript by Park and colleagues constitutes an important milestone in the analysis of OTX2 secretion, possibly valid for most HPs since the authors demonstrate that it applies to the homeodomain (DNA-binding domain), a structure highly conserved between all HPs and containing two signal peptides necessary for secretion and internalization, respectively. The main finding is that it is nuclear OTX2 which is secreted through the formulation of double-layered vesicles encompassing the inner and the outer nuclear membrane. The demonstration based on immunocytochemistry, electron microscopy, plus pharmacological and genetic approaches is solid, verified in vitro in transfected cells but also in vivo with a convincing physiological output. All in all, this is a timely and well-done study that deserves rapid publication.

In spite of this positive evaluation, there are some issues that need to be addressed, either experimentally or at an editorial level. They are presented below:

1. Unconventional secretion is not a HP privilege and the reader would certainly benefit from a comparison with what is known for molecules like IL1, FGF2, and a few others also secreted by still mysterious pathways. The comparison with the work done by other groups on ENGRAILED secretion, in particular its association with "caveolae-like vesicles", the necessity of a passage through the nucleus and the function of PIP2 would have been interesting, even if not mandatory. If the authors wish, this can be introduced in the discussion, in addition to the interesting comparison, in an evolutionary perspective, between yeast nuclear to vacuole vesicle transport and OTX2 nuclear "budding".

References of interest for HP secretion:

<https://doi.org/10.1242/jcs.244327>

DOI: 10.1242/dev.124.10.1865

DOI:<https://doi.org/10.1074/jbc.M609246200>

DOI: 10.1242/dev.129.15.3545

DOI: 10.1242/dev.126.14.3183

DOI:[https://doi.org/10.1016/S0960-9822\(07\)00346-6](https://doi.org/10.1016/S0960-9822(07)00346-6)

2. There is page 5 line 102 a sentence on non-secretory HPs. Indeed, secretion has been studied in a physiological context for about 10 HPs only. But the same group has reported that out of 160 tested 150 HPS can transfer in vitro and in vivo (DOI:<https://doi.org/10.1016/j.celrep.2019.06.056>). The same sentence qualifies HOXD4 as a non-secreted HP in spite of the presence of the two sequences which in the homeodomain are necessary and sufficient for secretion and internalization. In fact, when secretion is studied in a non-physiological context, one must be aware of the fact that the cells which are transfected might not allow secretion, for example through post-translational modifications that block nuclear import (see references above).

3. Even tagged, secreted OTX2 can be reinternalized by all cells, even the producing ones (unless the tag precludes it). Recaptured OTX2 can gain direct access to the cytoplasm but one cannot totally eliminate the possibility that some of it is endocytosed (the tag could even increase this endocytosed OTX2 pool), followed by lysosome addressing. It might thus be useful to verify if part of the protein present in vesicles does not gain access to them through endocytosis.

4. The term of OTX2 aggregates or insoluble OTX2 is confusing. It is not always clear whether one is talking of insoluble aggregates or of vesicular OTX2 that pellets at high speed. This is true for intracellular an extracellular OTX2. In the latter case, even if it is well-established that some OTX2 is free in the medium, the observation raises the possibility of an association with extracellular vesicles. Furthermore, an association with vesicles does not necessarily mean totally intravesicular. The protein

could be in the vesicle lumen or bound to its surface. It might even shuttle between the surface and the lumen. Simple experiments such as protection against proteolysis or vesicle treatment with high salt, or sodium carbonate at pH9, might help resolving this ambiguity.

5. In the *in vivo* pharmacological treatment, or in the Tor1a Δ mutant, it is possible that many proteins with unconventional secretion, including a fraction of the 300 HPs, or so, expressed in the mouse are affected. It is thus quite surprising that the authors did not detect morphological or physiological defects, beyond a decrease in visual acuity. Conversely, whether one can attribute the decrease in visual acuity only to non-cell autonomous OTX2 partial loss of function is debatable. In this visual acuity analysis, the reviewer did not find the information on the age at which the experiment was done. Given that the critical period lasts between P20 and P40 in V1, this is an important piece of information.

6. Although it is clear that OTX2 secretion is, in part, regulated by the nuclear "budding", what happens after is not limpid. Do the vesicles travel to the plasma membrane and fuse with it? Is there a part of lysosomes that fuse with the membrane? Are extracellular vesicles released in the medium? Here again, the localization of OTX2 within the vesicles or at their surface represents an important information for the understanding of its secretion pathway.

7. It is proposed that inhibiting OTX2 nuclear export induces its nuclear aggregation with "pathological" consequences, including apoptosis (not shown is not acceptable). The reviewer is not convinced that this is an important aspect of the manuscript and believes, wrong or right, that it should be part of another study. He/she is even less convinced by the poly Q or poly A hypothesis. Not that this might not be true, but aggregation of poly-Q containing proteins requires more than 5 Q residues. For example, in many poly-Q diseases triplet amplification must reach a much higher level (beyond 37 for huntingtin).

8. This is minor, but figure legends should include a definition of all abbreviations used, which is not the case.

Again, and in spite of the latter comments that deserve answers, the reviewer considers that the study is important and should be made public without excessive delay.

Reviewer #2 (Remarks to the Author):

The study by Park et al. aims at deciphering the mechanism of OTX2 homeoprotein unconventional secretion pathway. Secretion of this nuclear transcription factor coupled to its internalization into target neurons (pv cells) permits its action as a paracrine factor that regulates plasticity of the mouse visual cortex. From, microscopy and biochemical approaches, OTX2 is detected in cytoplasmic vesicles enriched in nuclear envelope and lysosome markers suggesting their nuclear origin and their subsequent addressing to lysosomal compartments. The role of this vesicles in OTX2 secretion is supported by the analysis of a non-nuclear mutant of OTX2 and by pharmacological inhibition of lysosomal degradation leading to a decrease and increase of secretion respectively.

This study is an important and original piece of work gathering multiple approaches applied to *in vitro* and *in vivo* models. The two main finding of this study are the discovery of a new mechanism of nuclear egress that, to my knowledge has never been reported, and its involvement in the subsequent unconventional secretion of OTX2, which is indispensable to the physiological function of this protein. One of the most surprising results of this study is the requirement of a double layered membrane structure of cytoplasmic vesicles. Relying on similarity with the atypical nuclear export mechanism of HSV and RNPs, the authors demonstrate that biogenesis of these vesicles require the coordinate activities of TOR1A ATPase and dynamin. Importantly, lowering Tor1A activity in cells that secrete OTX2, reduces both OTX2 intercellular transfer and action on plasticity. The authors next propose that

OTX2 interaction with the LINC complex that bridge inner and outer nuclear envelope membranes is required for nuclear egress and inhibition of this bridge impedes OTX2 secretion in vitro and paracrine action in vivo. An additional function of OTX2 nuclear egress is to prevent its aggregation in the nucleus

Although I acknowledge the overall quality and originality of this study, some points would need further validation, in particular the first one that to my mind result from a misinterpretation of the technique used.

1

Interaction with the LINC complex is central in the proposed mechanism of OTX2 nuclear egress but unless I missed some important point, the strategy used to demonstrate this interaction is not adapted. If the authors refer to the split GFP technique described in Cabantous et al. (<https://doi.org/10.1038/nbt1044>, no reference is provided in the text), it is based on bimolecular complementation and not on FRET. In the original report, the two GFP fragments (GFP1_10 and GFP11) spontaneously assemble, meaning that GFP fluorescence is not a reliable measure of interaction but only of co-localisation that furthermore, can be promoted by the GFP moiety. Interaction should be demonstrated by other means (e.g. Co-IP, actual FRET). It raises another concern in the experiment Fig5d. Once assembled, the high stability of the interaction of the 2 GFP fragments (Cabantous et al.) would promote its entrapment at the nuclear envelope membrane, thus restricting its secretion and not increasing as it is observed.

This question is of importance as, to my knowledge, nuclear egress processes involve a de-envelopment step at the outer nuclear envelope membrane which invariably lead to the release of membrane-free material. How OTX2 would use (induce?) a different route must be discussed.

2

Identification of the OTX2 cytoplasmic vesicles relies on 3 complementary approaches, fluorescent microscopy, biochemical fractionation and electronic microscopy. The distribution of the different markers in the gradient fig 1d strongly indicate that the 7-9 fractions containing OTX2 are composed of heterogeneous populations of vesicles. Of note, another secreted homeoprotein, engrailed-2, is known to be detected in caveolae and membrane rafts (10.1242/dev.124.10.1865), enriched in flotillin1. The gradient could be adjusted to better separate these fractions but it might turn to be uneasy. A better option would be to analyse how the distribution of OTX2 and the markers is altered upon modulation of OTX2 nuclear egress and/or secretion (OTX2 mutant, TOR1AΔE expression, Bafilomycin, EGFP-KASH2) as they might act on different populations of vesicles. In the same line, a semi quantitative analysis of the co-localisation data (Fig 1f) would be useful as the markers might not be present in all vesicles.

3

If OTX2 nuclear egress is convincingly demonstrated, the involvement of lysosomes in the subsequent secretion of these vesicles is less evident. Beside acting on lysosomes, Bafilomycin is one of the most potent inducers of exosome secretion. If secreted OTX2 arises from double-layered membrane vesicles, how can it be released free in the medium and not sequestered into vesicles (fig2f,g). The involvement of vesicle membrane shedding by lipase (in lysosomes?, in the medium?) proposed by the authors is mainly hypothetical and could be presented as such.

Additional points

The role of heparin treatment cannot be evaluated as OTX2 signal seems to have been inadvertently duplicated in ctrl and heparin treatment (Fig 1d).

Detection of detergent insoluble OTX2 in Fig7c,d and extended Fig 7 b,c is far from convincing, especially when compared to the amount recovered in the soluble fraction. From a more general point of view, the space devoted to OTX2 aggregation in the discussion seems to me disproportionate unless the authors estimate that the main function of OTX2 nuclear egress is to clear the nucleus from insoluble OTX2 but not to promote its secretion

The in vivo modulation of OTX2 paracrine activity convincingly correlates with

the proposed model of OTX2 secretion. The putative role of moderate degeneration of ChP cells in Tor1a E/+ expression on the decrease of OTX2 secretion could not be formally ruled-out, it would be interesting to analyse apoptosis in the GFP-KASH expressing mice.

Why is the size of OTX2 shifted in the soluble cytoplasmic fraction in fig 1c but not in 1d

Alain Joliot

Reviewer #3 (Remarks to the Author):

These authors present a series of experiments in support of the hypothesis that Otx2 (and perhaps other secreted HPs) exit the nucleus through a nuclear membrane-based budding process involving torsin and LINC complex. The data in general are of high quality and incorporating in vitro and in vivo studies in service of their study is a strength as well. Many of the observations will be of significant interest to many as this type of nuclear membrane-based pathway remains poorly understood and indeed controversial. That said, the data presented are not convincing in demonstrating conclusively that Otx2 is indeed transported across the nuclear membrane via a budding process, and the in vivo data, while interesting and theoretically consistent with the possibility, also do not demonstrate this in any way but instead are predicted consequences of an unproven mechanism. Notably, all the data regarding Otx2 in the first 3 figures is based on overexpression with no attempt to examine endogenous protein. In Fig 4f they examine "Otx2 containing vesicles" but never really demonstrate such "vesicles" in any conclusive way (the IF in 4e is entirely unconvincing). Figures 3a and 4b, both EM, are pseudocolored in a way that advances the authors conclusions about the data, but are really guesses about what might be going on. In 3a there is nothing even remotely like a proposed NE bud and in 4b it is far from clear that these standard EM images represent what the author claims. Methods such as serial EM tomography with reconstruction – at a minimum – are needed to make any reliable claims about the origin and topography of the structures and ideally this would include immuno of endogenous protein.

Other comments:

1. Overexpression of any torsin proteins leads to nuclear membrane dysmorphogenesis that has nonspecific effects. This is not accounted for in the experiments.
2. If Otx2 is getting sequestered in the CP and is low in CSF, why are levels not increased in the CP cells (4g,h)
3. How were the PV neuron and other counts done – nothing is described in the methods. Were these counts blinded? Were they done using stereological methods?
4. How is "visual acuity" measured – not in methods.
5. All of the findings in KASH etc mutants are of interest but authors should not claim they prove anything about fundamental claim regarding nuclear egress unless they show that specific data - not simply proposed downstream consequences.
6. Why do the authors believe they are seeing Nuclear membrane abnormalities when these have been looked for and specifically stated to be absent in heterozygous torsinA mutant mice?
7. The authors should omit any mention of a "dominant negative" effect of torsin unless they can demonstrate. As stated, this is speculation.

Reviewers' comments (11 font, bold-underlined parts are the conclusive remarks)
Authors' responses to the comments (12 font, italic)

Reviewer #1

Although homeoprotein (HP) transfer is a now well-established signaling pathway, the unconventional mechanisms for HP internalization and secretion are yet to be fully understood. **The manuscript by Park and colleagues constitutes an important milestone in the analysis of OTX2 secretion**, possibly valid for most HPs since the authors demonstrate that it applies to the homeodomain (DNA-binding domain), a structure highly conserved between all HPs and containing two signal peptides necessary for secretion and internalization, respectively. The main finding is that it is nuclear OTX2 which is secreted through the formulation of double-layered vesicles encompassing the inner and the outer nuclear membrane. The demonstration based on immunocytochemistry, electron microscopy, plus pharmacological and genetic approaches is solid, verified in vitro in transfected cells but also in vivo with a convincing physiological output. **All in all, this is a timely and well-done study that deserves rapid publication.**

In spite of this positive evaluation, there are some issues that need to be addressed, either experimentally or at an editorial level. They are presented below:

1. Unconventional secretion is not a HP privilege and the reader would certainly benefit from a comparison with what is known for molecules like IL1, FGF2, and a few others also secreted by still mysterious pathways. The comparison with the work done by other groups on ENGRAILED secretion, in particular its association with "**caveolae-like vesicles**", the necessity of a passage through the nucleus and the function of **PIP2** would have been interesting, even if not mandatory. If the authors wish, this **can be introduced in the discussion**, in addition to the interesting comparison, in an evolutionary perspective, between **yeast nuclear to vacuole vesicle transport and OTX2 nuclear "budding"**.

References of interest for HP secretion:

<https://doi.org/10.1242/jcs.244327>

DOI: 10.1242/dev.124.10.1865

DOI:<https://doi.org/10.1074/jbc.M609246200>

DOI: 10.1242/dev.129.15.3545

DOI: 10.1242/dev.126.14.3183

DOI:[https://doi.org/10.1016/S0960-9822\(07\)00346-6](https://doi.org/10.1016/S0960-9822(07)00346-6)

: Thank you for these comments. On page 19, we now discuss the possibility that HPs could exit the nucleus through binding to PIP₂ in the inner leaflet of the nuclear inner membrane and the caveolae. We also added several of the above-listed references, selected because we could not add all of the recommended references owing to the spatial limitation.

2. There is page 5 line 102 a sentence on non-secretory HPs. Indeed, secretion has been studied in a physiological context for about 10 HPs only. But the same group has reported that out of 160 tested 150 HPs can transfer in vitro and in vivo (DOI:<https://doi.org/10.1016/j.celrep.2019.06.056>). The same sentence qualifies HOXD4 as a non-secreted HP in spite of the presence of the two sequences which in the homeodomain are necessary and sufficient for secretion and internalization. In fact, when **secretion is studied in a non-physiological context**, one must be aware of the fact that the cells which are transfected might not allow secretion, for example through post-translational modifications that block nuclear import (see references above).

: We agree that the cell lines used for screening might not provide an appropriate cellular context that supports the secretion of HOXD4 and SHOX2. Thus, we concluded that HP

could be a cell context-dependent event (Lee et al., 2019).

3. Even tagged, secreted OTX2 can be reinternalized by all cells, even the producing ones (unless the tag precludes it). Recaptured OTX2 can gain direct access to the cytoplasm but one cannot totally eliminate the possibility that some of it is endocytosed (the tag could even increase this endocytosed OTX2 pool), followed by lysosome addressing. **It might thus be useful to verify if part of the protein present in vesicles does not gain access to them through endocytosis.**

: We did not find OTX2 in the endosomes (Fig. 1f and g). However, our original work did not rule out the possibility that the increase of cytoplasmic OTX2 in BafA1-treated cells resulted from an increase of endocytosed OTX2. To test this hypothesis, we added recombinant OTX2 to the growth media of vehicle- or BafA1-treated HeLa cells and detected OTX2 in the growth medium and intracellular space by DB and WB analyses, respectively. The results show that the amounts of extracellular and intracellular OTX2 were insensitive to BafA1 treatment, suggesting that the increase was unlikely to reflect endocytosed OTX2. We provide the following results for the reviewers' inspection only.

4. The term of OTX2 aggregates or insoluble OTX2 is confusing. It is not always clear whether **one is talking of insoluble aggregates or of vesicular OTX2 that pellets at high speed.** This is true for intracellular and extracellular OTX2.

: Small vesicles cannot be precipitated under the centrifugation conditions we used to separate soluble and insoluble OTX2 (please see the details in the Methods). Therefore, the OTX2 in the precipitates was unlikely to reflect its presence in trafficking membrane vesicles, but rather that in large cytoplasmic aggregates or intracellular structures, such as nuclei and mitochondria.

In the latter case, even if it is well-established that some OTX2 is free in the medium, the observation raises **the possibility of an association with extracellular vesicles.** Furthermore, an association with vesicles does not necessarily mean totally intravesicular. The protein could be in the vesicle lumen or **bound to its surface.** It might even shuttle between the surface and the lumen. **Simple experiments such as protection against proteolysis or vesicle treatment with high salt, or sodium carbonate at pH9, might help resolving this ambiguity.**

: We agree that OTX2 could bind the surface of membrane vesicles inside and outside of the cell, because it has affinity to negatively charged molecules, such as sugars and lipid head groups.

To address the reviewer's comment, we investigated whether OTX2 binds to the

surface of extracellular vesicles. To detach proteins bound the surface of extracellular vesicles, we added NaCl (0.5 M, final) to growth medium, conditioned by OTX2-expressing cells, prior to the step during which membrane vesicles (P100 fraction) were separated from the medium (S100 fraction) by ultracentrifugation. NaCl treatment decreased the amount of OTX2 in the P100 fraction but elevated that in the S100 fraction, suggesting that significant amounts of extracellular OTX2 attached to the outer leaflets of the vesicles.

To assess proteins inside the extracellular membrane vesicles, we also treated growth medium conditioned by OTX2-expressing cells with proteinase-K (PK) in the presence or absence of sodium dodecyl sulfate (SDS), which disrupts membrane vesicles. The level of OTX2 was significantly decreased by PK treatment (to 40% of untreated samples) and disappeared almost completely (<10% of untreated samples) under treatment with PK in the presence of SDS. These results show that the majority (>50%) of extracellular OTX2 is present in the soluble fraction or at the outer leaflets of the vesicles. We provide the results below only for the reviewers' inspection.

Identification of physical status of OTX2 in the growth medium. **a**, Growth medium was collected from HeLa cells overexpressing V5-OTX2 together with GFP and treated with sodium chloride (NaCl, 0.5M) or vehicle (same volume of fresh growth medium) for 1 hour at 4°C. The extracellular vesicles in the precipitate (P100) fraction were separated from the supernatant (S100) fraction by ultracentrifugation. V5-OTX2 and GFP in the S100 and P100 fractions were then examined by DB and WB, respectively. **b**, The graph shows relative intensity of dots and WB bands corresponding to V5-OTX2 in the S100 and P100 fractions. The columns represent means and error bars denote SEM. Results were obtained from six independent experiments (*, $p < 0.05$; ****, $p < 0.0001$; Student's *t*-test). **c**, Alternatively, the growth medium was treated with proteinase K (PK, 1 mg/ml), sodium dodecyl sulfate (SDS, 0.1%), or vehicle (medium) for 1 hour at 4°C prior to the precipitation of proteins in the growth medium by trichloroacetic acid (TCA). V5-OTX2 and GFP in the precipitates were then detected by WB. **d**, The graph shows the relative intensity of WB bands for OTX2. The columns represent means and error bars denote SEM. Results were obtained from five independent experiments (****, $p < 0.0001$; ANOVA).

5. In the in vivo pharmacological treatment, or in the Tor1aΔ mutant, it is possible that many proteins with unconventional secretion, including a fraction of the 300 HPs, or so, expressed in the mouse are affected. It is thus quite surprising that the authors did not detect morphological or physiological defects, beyond a decrease in visual acuity. Conversely, **whether one can attribute the decrease in visual acuity only to non-cell autonomous OTX2 partial loss of function is debatable**. In this visual acuity analysis, **the reviewer did not find the information on the age at which the experiment was done**. Given that the critical period lasts between P20 and P40 in V1, this is an important piece of information.

: We monitored the visual acuity of mice at P30, when the critical period is closed by the transfer of Otx2 to PV cells in the V1 cortex. We also provided information on mouse

ages in the figure legends.

Tor1a^{ΔE/+} mice exhibit torsion dystonia in a mouse strain-specific manner, whereas Tor1a^{ΔE/ΔE} mice fail to survive beyond two weeks in all strains (Tanabe et al., 2012). The C57BL/6 Tor1a^{ΔE/+} mice used in this study did not show remarkable degenerative phenotypes by 12 months strains, whereas the majority 129S6/SvEvTac Tor1a^{ΔE/+} mice died within a week after birth (Tanabe et al., 2012). Thus, we did not see significant morphological or physiological defects in the tested mice by 12 months of age.

6. Although it is clear that OTX2 secretion is, in part, regulated by the nuclear “budding”, what happens after is not limp. **Do the vesicles travel to the plasma membrane and fuse with it?**
: We cannot rule out the possibility that cytoplasmic OTX2-containing vesicles are secreted by fusing directly to the plasma membrane. For the double-layered nuclear budding vesicles, fusion of the outer membrane to plasma membranes might release OTX2 in the inner membrane vesicles to the extracellular space. However, based on our observation that extracellular OTX2 is mostly naked or bound to the surface of extracellular vesicles (please see the data provided in our response to comment #4 of this reviewer), we believe that this route is unlikely to be the major pathway.

Is there a part of lysosomes that fuse with the membrane?

: The fusion of lysosomes to the plasma membrane has been well reported (Luzio et al., 2007; Reddy et al., 2001; Rodriguez et al., 1997). However, it is not known whether a certain part of the lysosome selectively fuses to the plasma membrane.

Are extracellular vesicles released in the medium?

: In the case of double-layered vesicles, fusion of the outer membrane to lysosomes and plasma membranes might release OTX2-containing inner membrane vesicles into the lysosome and the extracellular space, respectively. Given the increase of OTX2 in cells treated with lysosomal inhibitors (i.e., BafA1 and CQ), we believe that OTX2 is more likely to be transported to the lysosome for degradation. Therefore, the inner membrane vesicles containing OTX2 would be decapsulated by lysosomal lipases to expose OTX2 for lysosomal proteolytic degradation (Supplementary Fig. 4). Some OTX2 that survives the lysosomal lipases and proteases might be released to the extracellular space from the single-layer vesicles and in naked form, respectively, upon fusion of the lysosome to the plasma membrane.

Here again, the localization of OTX2 within the vesicles or at their surface represents an important information for the understanding of its secretion pathway.

: Please see the data provided in our response to comment #4 of this reviewer.

7. It is proposed that inhibiting OTX2 nuclear export induces its nuclear aggregation with “pathological” consequences, including apoptosis (not shown is not acceptable). The reviewer is not convinced that this is an important aspect of the manuscript and believes, wrong or right, that it should be part of another study. **He/she is even less convinced by the poly Q or poly A hypothesis.** Not that this might not be true, but aggregation of poly-Q containing proteins requires more than 5 Q residues. For example, in many poly-Q diseases triplet amplification must reach a much higher level (beyond 37 for huntingtin).

: To comply with the reviewer’s suggestion, we removed the OTX2-polyQ-related results from the revised version and will publish it in another report after further integration of physiological evidence.

8. This is minor, but figure legends should include a definition of all abbreviations used, which is not the case.

: We collected all abbreviations and now provide them in a separate footnote.

Again, and in spite of the latter comments that deserve answers, **the reviewer considers that the study is important and should be made public without excessive delay.**

Reviewer #2

The study by Park et al. aims at deciphering the mechanism of OTX2 homeoprotein unconventional secretion pathway. Secretion of this nuclear transcription factor coupled to its internalization into target neurons (pv cells) permits its action as a paracrine factor that regulates plasticity of the mouse visual cortex. From, microscopy and biochemical approaches, OTX2 is detected in cytoplasmic vesicles enriched in nuclear envelope and lysosome markers suggesting their nuclear origin and their subsequent addressing to lysosomal compartments. The role of this vesicles in OTX2 secretion is supported by the analysis of a non-nuclear mutant of OTX2 and by pharmacological inhibition of lysosomal degradation leading to a decrease and increase of secretion respectively.

This study is an important and original piece of work gathering multiple approaches applied to in vitro and in vivo models. The two main findings of this study are the discovery of a new mechanism of nuclear egress that, to my knowledge has never been reported, and its involvement in the subsequent unconventional secretion of OTX2, which is indispensable to the physiological function of this protein.

One of the most surprising results of this study is the requirement of a double layered membrane structure of cytoplasmic vesicles. Relying on similarity with the atypical nuclear export mechanism of HSV and RNPs, the authors demonstrate that biogenesis of these vesicles require the coordinate activities of TOR1A ATPase and dynamin. Importantly, lowering Tor1A activity in cells that secrete OTX2, reduces both OTX2 intercellular transfer and action on plasticity. The authors next propose that OTX2 interaction with the LINC complex that bridge inner and outer nuclear envelope membranes is required for nuclear egress and inhibition of this bridge impedes OTX2 secretion in vitro and paracrine action in vivo. An additional function of OTX2 nuclear egress is to prevent its aggregation in the nucleus

Although I acknowledge the overall quality and originality of this study, some points would need further validation, in particular the first one that to my mind result from a misinterpretation of the technique used.

1. Interaction with the LINC complex is central in the proposed mechanism of OTX2 nuclear egress but unless I missed some important point, the strategy used to demonstrate this interaction is not adapted. If the authors refer to the split GFP technique described in Cabantous et al. (<https://doi.org/10.1038/nbt1044>, no reference is provided in the text), it is based on bimolecular complementation and not on FRET. In the original report, **the two GFP fragments (GFP1_10 and GFP11) spontaneously assemble, meaning that GFP fluorescence is not a reliable measure of interaction but only of co-localisation** that furthermore, can be promoted by the GFP moiety. **Interaction should be demonstrated by other means (e.g. Co-IP, actual FRET).**

: We respect the reviewer's criticism regarding the split GFP system. As the reviewer indicates, the N-terminal and C-terminal fragments of GFP interact readily when these have a physical proximity. However, in our experimental condition, the fluorescence signals were not detectable from the cells co-expressing those two fragments unless we intensify the signals. Moreover, the GFP signal increased significantly when the N-terminal fragment fused to OTX2 was co-expressed with the C-terminal fragment connected to SUN1 and LAP1, whereas the signals were not elevated when the N-terminal fragment fused to OTX2 was co-expressed with the C-terminal fragment connected to LMNA1 and SYNE2. We interpreted this as indicating that a physical interaction between OTX2 and SUN1 or LAP1 might enhance the interaction between the GFP fragments.

Nevertheless, as the reviewer notes, this is not a clean system that can show physical interaction between two proteins. We thus tried to find evidence of protein interaction by co-immunoprecipitation experiments. Our results presented in revised Fig.

6b show that OTX2 strongly co-precipitated with SUN1 and LAP1 and weakly co-precipitated with LMNA1. Interestingly, SYNE2, which might not have direct access to OTX2 and failed to enhance the split GFP signal, also co-immunoprecipitated with OTX2, suggesting that SYNE2 might be co-precipitated with OTX2 via SUN1.

We further found that SUN1 and LAP1 increased the numbers of cytoplasmic OTX2 puncta (revised Fig. 6d,e), which was similar to the results obtained by the split GFP system. These results suggest that SUN1 and LAP1 might interact directly with OTX2 to trigger the budding of OTX2-containing nuclear vesicles to the cytoplasm.

It raises another concern in the experiment Fig5d. Once assembled, **the high stability of the interaction of the 2 GFP fragments (Cabantous et al.) would promote its entrapment at the nuclear envelope membrane, thus restricting its secretion and not increasing as it is observed.**

: Our original data showed that co-expression of SUN1-GFP11 or LAP1-GFP11 promoted the secretion of OTX2-GFP1-10. Therefore, the potential entrapment of OTX2 in the nuclear envelope by the spontaneously assembled GFP fragments might not restrict secretion, but rather promote it. For this revision, we tested the secretion of OTX2 in cells co-expressing LINC components without the split GFP system. We still observed that secretion of OTX2 was elevated by SUN1 and LAP1 (revised Fig. 6f).

This question is of importance as, to my knowledge, **nuclear egress processes involve a de-envelopment step at the outer nuclear envelope membrane which invariably lead to the release of membrane-free material.** How OTX2 would use (induce?) a different route must be discussed.

: We propose that the nuclear egress of OTX2 (and perhaps other HPs) might differ from that of HSV and RNP, which are trapped in the inner membrane vesicles that fuse to the outer nuclear membrane to release their contents into the cytoplasm. To form nuclear budding vesicles, HSV uses pUL31 to bind pUL34, which integrates to the nuclear inner membrane and deforms the nuclear lamina and LINC complex (Maric et al., 2011; Mettenleiter et al., 2006). On the contrary, OTX2 mediates the LINC complex for nuclear egress. Consequently, nuclear OTX2 could be released into the cytoplasm from double-layer membrane vesicles supported by the LINC complex, whereas the nuclear HSV capsids are released in the cytoplasm by fusion of the inner nuclear membrane vesicles to the outer nuclear membrane. However, given the presence of naked OTX2 in the cytoplasm (Fig. 3a,b,f,g), we do not rule out the possibility of de-envelopment at the outer nuclear membrane.

2. Identification of the OTX2 cytoplasmic vesicles relies on 3 complementary approaches, fluorescent microscopy, biochemical fractionation and electronic microscopy. The distribution of the different markers in the gradient fig 1d strongly indicate that the 7-9 fractions containing OTX2 are composed of heterogeneous populations of vesicles. Of note, another secreted homeoprotein, engrailed-2, is known to be detected in caveolae and membrane rafts (10.1242/dev.124.10.1865), enriched in flotillin1. The gradient could be adjusted to better separate these fractions but it might turn to be uneasy. **A better option would be to analyse how the distribution of OTX2 and the markers is altered upon modulation of OTX2 nuclear egress and/or secretion (OTX2 mutant, TOR1A Δ E expression, Bafilomycin, EGFP-KASH2)** as they might act on different populations of vesicles.

: We detected OTX2 in the nuclear and cytoplasmic fractions of cells expressing TOR1A Δ E and EGFP-KASH2 to determine whether these proteins could affect the transport of OTX2 from the nucleus to the cytoplasm. We found that the amount of OTX2 in the cytoplasmic fraction was decreased by co-expression of TOR1A Δ E or EGFP-KASH2. Together with our observation that the number of cytoplasmic OTX2 puncta was

reduced in cells expressing TOR1A Δ E and EGFP-KASH2, these results suggest that TOR1A Δ E and EGFP-KASH2 might decrease the cytoplasmic OTX2 level by suppressing the formation of nuclear membrane vesicles.

When seeking to examine the influence of TOR1A Δ E or EGFP-KASH2 on the distribution of OTX2 in the cytoplasmic compartments, FACS should be used to collect only cells co-expressing these proteins from among all cells on a culture dish, prior to sucrose gradient analysis. Intracellular protein distribution could be changed during FACS procedures. Thus, we decided not to examine the distribution of cytoplasmic OTX2 in those TOR1A Δ E- or EGFP-KASH2-coexpressing cells using a sucrose density gradient. Instead, we addressed the reviewer's suggestion by using sucrose gradient analysis to examine the fractional distribution of OTX2 and subcellular organelle markers in cells treated with BafA1, which does not need FACS. The results presented in Supplementary Fig. 3 show that BafA1 treatment did not change the distribution of OTX2 in the cytoplasm, but rather it increased the OTX2 level in LAMP2-enriched lysosomal fractions.

In the same line, **a semi quantitative analysis of the co-localisation data (Fig 1f) would be useful as the markers might not be present in all vesicles.**

: We quantified fluorescent signals of OTX2 that co-localized with intracellular organelle markers from among total cytoplasmic OTX2 signals, and provide the results in Fig. 1g.

3. If OTX2 nuclear egress is convincingly demonstrated, the involvement of lysosomes in the subsequent secretion of these vesicles is less evident. Beside acting on lysosomes, Bafilomycin is one of the most potent inducers of **exosome secretion**. If secreted OTX2 arises from double-layered membrane vesicles, how can it be released free in the medium and not sequestered into vesicles (fig2f,g). **The involvement of vesicle membrane shedding by lipase (in lysosomes?, in the medium?) proposed by the authors is mainly hypothetical and could be presented as such.**

: We treated cells with IMP and L1 (lysosomal lipase inhibitors) and detected OTX2 in the growth media and cell lysates (Supplementary Fig. 4). This manipulation increased OTX2 in the media and cytoplasm, but did not change the OTX2 level in the nucleus. These results suggest that lysosomal lipases are necessary for the shedding of OTX2-containing vesicles in the lysosome and/or extracellular space.

Additional points

The role of heparin treatment cannot be evaluated as OTX2 signal seems to have been inadvertently duplicated in ctrl and heparin treatment (Fig 1d).

: We apologize for the duplication of the OTX2 WB images. We corrected the results in the figure.

Detection of detergent insoluble OTX2 in Fig7c,d and extended Fig 7 b,c is far from convincing, especially when compared to the amount recovered in the soluble fraction. From a more general point of view, **the space devoted to OTX2 aggregation in the discussion seems to me disproportionate** unless the authors estimate that the main function of OTX2 nuclear egress is to clear the nucleus from insoluble OTX2 but not to promote its secretion

: In the revised manuscript, we removed the part discussing OTX2 aggregation by respecting the reviewer's suggestion. In addition, in response to the suggestions of reviewers #1 and #2, we also removed the OTX2-polyQ-related results from the revised version and will publish it in another report after the integration of physiological evidence. However, we think it is still necessary to provide our readers with our interpretation of the

physiological roles of this unconventional OTX2 nuclear egress. Therefore, we decided to keep Fig. 8 (original Fig. 7) in the revised manuscript.

The in vivo modulation of OTX2 paracrine activity convincingly correlates with the proposed model of OTX2 secretion. The putative role of moderate degeneration of ChP cells in Tor1a E/+ expression on the decrease of OTX2 secretion could not be formally ruled-out, it would be interesting to analyse apoptosis in the GFP-KASH expressing mice.

: We detected a few Casp3-positive apoptotic cells in ChP cells of LSL-KASH2;FoxJ1-CreER mice but not in ChP cells of FoxJ1-CreER littermates, which was similar to the results obtained in Tor1a $\Delta E/+$ mice. We provide the results below for the reviewers' inspection only.

Why is the size of OTX2 shifted in the soluble cytoplasmic fraction in fig 1c but not in 1d
: We do not yet clearly understand the mechanism underlying the differential electrophoretic mobilities of cytoplasmic OTX2. It is possible that OTX2 could be modified post-translationally by phosphorylation, acetylation, methylation, SUMOylation, or ubiquitination in the nucleus or vesicles. This might increase the molecular weight of the cytoplasmic OTX2, slowing its in-gel migration. Identifying the post-translational modification(s) should be a focus of future studies.

Reviewer #3

These authors present a series of experiments in support of the hypothesis that Otx2 (and perhaps other secreted HPs) exit the nucleus through a nuclear membrane-based budding process involving torsin and LINC complex. **The data in general are of high quality and incorporating in vitro and in vivo studies in service of their study is a strength as well. Many of the observations will be of significant interest to many as this type of nuclear membrane-based pathway remains poorly understood and indeed controversial.**

That said, the data presented are not convincing in demonstrating conclusively that Otx2 is indeed transported across the nuclear membrane via a budding process, and the in vivo data, while interesting and theoretically consistent with the possibility, also do not demonstrate this in any way but instead are predicted consequences of an unproven mechanism. Notably, **all the data regarding Otx2 in the first 3 figures is based on overexpression with no attempt to examine endogenous protein.** In Fig 4f they examine “Otx2 containing vesicles” but never really demonstrate such “vesicles” in any conclusive way (**the IF in 4e is entirely unconvincing**).

: We agree that the conclusion in our original manuscript was more or less based on the observations of overexpressed OTX2. For the revision, we sought to identify endogenous Otx2 in vivo by immuno-TEM (iTEM). We provide the iTEM results in revised Fig. 3f–h. These results show that significant amounts of cytoplasmic Otx2 are present in double-layered vesicles, which likely originated from nuclear membranes. We eliminated the confocal images presented in original Fig. 4e, since we now provide Otx2 iTEM data.

Figures 3a and 4b, both EM, are pseudocolored in a way that advances the authors conclusions about the data, but are really guesses about what might be going on. In **3a there is nothing even remotely like a proposed NE bud and in 4b it is far from clear that these standard EM images represent what the author claims. Methods such as serial EM tomography with reconstruction – at a minimum – are needed to make any reliable claims about the origin and topography of the structures** and ideally this would include immuno of endogenous protein.

: For the revision, we attempted to perform 3D reconstitution of Otx2-immunostained EM images by collaborating with EM specialists. Unfortunately, we failed to obtain 3D images of publishable grade, even after trying two different conditions. Further optimization of the procedures to conditions that can preserve membrane structures without sacrificing immunostaining quality must therefore await future studies. Instead, we added 2D iTEM images of Otx2 mouse ChP cells.

Other comments:

1. **Overexpression of any torsin proteins leads to nuclear membrane dysmorphogenesis** that has nonspecific effects. This is not accounted for in the experiments.

: We agree that there was dysmorphogenesis of the nuclear membranes in TOR1A- and TOR1AΔE-overexpressing cells. We added this interpretation to the Results section (page 11).

2. If Otx2 is getting sequestered in the CP and is low in CSF, **why are levels not increased in the CP cells (4g,h).**

: The amount of secreted Otx2 represents less than 5% that of nuclear Otx2 (Lee et al., 2019). Therefore, the failure of Otx2 secretion might not be sufficient to significantly increase the total amount of Otx2. We added this interpretation to the Discussion (page 18).

3. **How were the PV neuron and other counts done** – nothing is described in the methods. Were these counts blinded? Were they done using stereological methods?

: We added the utilized counting method to the Immunostaining section of the Methods (page 23).

4. **How is “visual acuity” measured** – not in methods.

: This information was provided on page 23 of the original manuscript. In the revised manuscript, it is presented on page 24.

5. All of the findings in KASH etc mutants are of interest but **authors should not claim they prove anything about fundamental claim regarding nuclear egress unless they show that specific data** – not simply proposed downstream consequences.

: We toned down this message in the revised text (page 13).

6. **Why do the authors believe they are seeing nuclear membrane abnormalities when these have been looked for** and specifically stated to be absent in heterozygous torsinA mutant mice?

: Tor1a^{ΔE/ΔE} mice fail to survive to birth (Tanabe et al., 2012), therefore we could not investigate the changes in the cells expressing homozygously. In addition, heterozygous mutation of Tor1a was found to be enough to change the membrane structure in other mouse cell types (Goodchild et al., 2005; Tanabe et al., 2016). We, thus, hypothesized that the nuclear membrane structure of Tor1a^{ΔE/+} mouse ChP cells might also be altered. This prompted us to investigate the structure by TEM. Indeed, we found that the inter-nuclear membrane spaces are enlarged in Tor1a^{ΔE/+} mouse ChP cells, in comparison with the space of wild-type ChP cells.

7. The authors **should omit any mention of a "dominant negative" effect of torsin** unless they can demonstrate. As stated, this is speculation.

: We changed this phrase to “inactive mutant” in the revised manuscript.

References

- Goodchild, R.E., Kim, C.E., and Dauer, W.T. (2005). Loss of the dystonia-associated protein torsinA selectively disrupts the neuronal nuclear envelope. *Neuron* *48*, 923-932.
- Luzio, J.P., Pryor, P.R., and Bright, N.A. (2007). Lysosomes: fusion and function. *Nat Rev Mol Cell Biol* *8*, 622-632.
- Reddy, A., Caler, E.V., and Andrews, N.W. (2001). Plasma membrane repair is mediated by Ca²⁺-regulated exocytosis of lysosomes. *Cell* *106*, 157-169.
- Rodriguez, A., Webster, P., Ortego, J., and Andrews, N.W. (1997). Lysosomes behave as Ca²⁺-regulated exocytic vesicles in fibroblasts and epithelial cells. *J Cell Biol* *137*, 93-104.
- Tanabe, L.M., Liang, C.C., and Dauer, W.T. (2016). Neuronal Nuclear Membrane Budding Occurs during a Developmental Window Modulated by Torsin Paralogs. *Cell Rep* *16*, 3322-3333.
- Tanabe, L.M., Martin, C., and Dauer, W.T. (2012). Genetic background modulates the phenotype of a mouse model of DYT1 dystonia. *PLoS One* *7*, e32245.

REVIEWERS' COMMENTS

Reviewer #1 (Remarks to the Author):

The main points raised in my first review have been addressed either at an editorial level or an experimental one. The conclusions are convincing and changes appropriate. I see no reason for retarding publication of this important study.

Reviewer #2 (Remarks to the Author):

The revised version have been significantly improved by the authors and answered most of my comments. I recommend its publication.